# Dataset Distillation as Pushforward Optimal Quantization

**Hong Ye Tan**
UCLA
University of Cambridge
hyt35@math.ucla.edu

**Emma Slade**
Tangram Therapeutics
GSK.ai
emma.slade@tangramtx.com

## ABSTRACT

Dataset distillation aims to find a small synthetic training set, such that training on the synthetic data achieves similar performance to training on a larger training dataset. Early methods solve this by interpreting the distillation problem as a bi-level optimization problem. On the other hand, disentangled methods bypass pixel-space optimization by matching data distributions and using generative techniques, leading to better computational complexity in terms of size of both training and distilled datasets. We demonstrate that by using latent spaces, the empirically successful disentangled methods can be reformulated as an optimal quantization problem, where a finite set of points is found to approximate the underlying probability measure. In particular, we link disentangled dataset distillation methods to the classical problem of optimal quantization, and are the first to demonstrate consistency of distilled datasets for diffusion-based generative priors. We propose Dataset Distillation by Optimal Quantization (DDOQ), based on clustering in the latent space of latent diffusion models. Compared to a similar clustering method $D^4M$, we achieve better performance and inter-model generalization on the ImageNet-1K dataset using the same model and with trivial additional computation, achieving SOTA performance in higher image-per-class settings. Using the distilled noise initializations in a stronger diffusion transformer model, we obtain competitive or SOTA distillation performance on ImageNet-1K and its subsets, outperforming recent diffusion guidance methods.

## 1 INTRODUCTION

Training powerful neural networks requires a large amount of data, and thus induces high computational requirements. *Dataset distillation* (DD) targets this computational difficulty by changing the data, as opposed to other parts of training such as optimization or architecture (Wang et al., 2018). The DD objective consists of finding a synthetic training set, such that training a neural network on the synthetic data yields similar performance.

There are several closely related notions of reducing computational load when training new models on datasets. Core-set methods find a subset of training data (as opposed to synthetic data) that achieve good training performance (Mirzasoleiman et al., 2020; Feldman, 2020). Model distillation, sometimes known as knowledge distillation, aims to train a smaller model that predicts the output of a larger model (Gou et al., 2021; Polino et al., 2018). Importance sampling methods accelerate training by weighting training data, finding examples that are more influential for training (Paul et al., 2021). For more detailed surveys on dataset distillation methods and techniques, we refer to (Yu et al., 2023; Sachdeva & McAuley, 2023).

### 1.1 BI-LEVEL FORMULATION OF DATASET DISTILLATION

Denote a training set (more generally, distribution of training data) by $\mathcal{T}$, and the expected and empirical risks (test and training loss) by $\mathcal{R}$ and $\mathcal{L}$ respectively, evaluated for some parameter $\theta$. The goal of DD is to find a synthetic dataset $\mathcal{S}$ (of given size) minimizing the test loss discrepancy

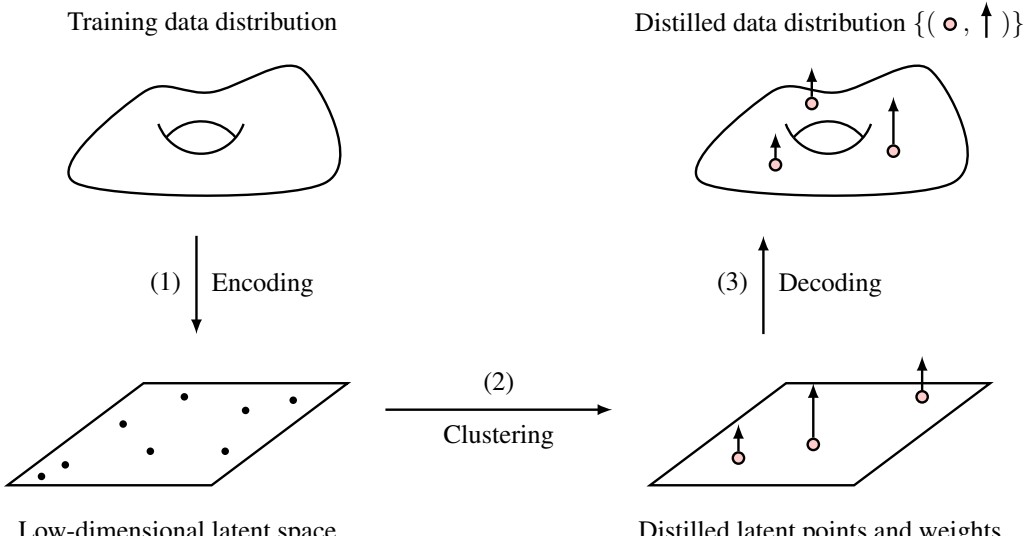

Figure 1: Sketch of the proposed method pipeline. Using an encoder/decoder model, we map our high dimensional data to a low-dimensional space, which is then clustered using $k$-means. The clustered latent points and weights are then decoded to obtain the distilled data. This work argues that the weights are important when decoding; furthermore, "disentangled" distillation using an encoder-cluster-decoder framework is asymptotically consistent.

(Sachdeva & McAuley, 2023):

$$\mathcal{S} = \arg\min_S \left| \mathcal{R}(\arg\min_\theta \mathcal{L}(S)) - \mathcal{R}(\arg\min_\theta \mathcal{L}(\mathcal{T})) \right|. \tag{1}$$

This formulation is computationally intractable. Approximations include replacing the minimum discrepancy objective with maximum test performance, replacing the learning algorithm $\Phi$ with an inner neural network optimization problem, and solving the outer minimization problem using gradient methods. Common heuristic relaxations to the bi-level formulation (1) include meta-learning (Wang et al., 2018; Deng & Russakovsky, 2022), distribution matching (Zhao & Bilen, 2023), and trajectory matching (Cazenavette et al., 2022). Other methods include neural feature matching (Zhou et al., 2022; Loo et al., 2022) and the corresponding neural tangent kernel methods (Nguyen et al., 2021; 2020), representative matching (Liu et al., 2023b), and group robustness (Vahidian et al., 2024). For better scaling, Cazenavette et al. (2022); Moser et al. (2024) consider using generative priors such as GANs to generate more visually coherent images, increasing performance and replacing the need for optimization with a neural network inversion task.

While the bi-level formulation follows naturally from the qualitative problem statement of dataset distillation, there are two main drawbacks, namely **computational complexity** and **model architecture dependence**. The dimensionality of the underlying optimization problems limit the applicability on large scale datasets, which are particularly useful for computationally limited applications. For example, the ImageNet-1K dataset consists of 1.2M training images, totalling over 120GB of memory (Deng et al., 2009). The full dataset ImageNet-21K consists of over 14M images and takes up around 1.2TB of memory, which is generally infeasible to train expert models on, and makes backpropagation through network training steps impossible.

## 1.2 DISENTANGLED AND DIFFUSION METHODS

Yin et al. (2023) is the first work to "disentangle" the bi-level optimization framework into three separate problems, named *Squeeze, Recover and Relabel* (SRe²L). In particular, the inner neural network optimization problem is replaced with matching statistics of batch-normalization layers. *Curriculum Data Augmentation* (CDA) uses adaptive training to get more performance (Yin & Shen, 2024). Liu et al. (2023a) considers optimizing images such that neural network features are close to Wasserstein barycenters of the training image features. Sun et al. (2024) proposes *Realistic Diverse*

*and Efficient Dataset Distillation* (RDED), which replaces the latent clustering objective with a patch-based adversarial objective. Su et al. (2024) considers clustering directly in the latent space of a latent diffusion model (LDM) (Rombach et al., 2022), named *Dataset Distillation via Disentangled Diffusion Model* ($D^4M$). This avoids backpropagation when distilling and has constant memory usage with respect to images per class (IPC), a direct advantage over the linear memory scaling of optimization-based methods. Recent state-of-the-art methods consider fine-tuning diffusion models to have better statistics (Gu et al., 2024), and guiding the diffusion to maximize the influence of the distilled points (Chen et al., 2025), similarly to active learning.

While dataset distillation has had extensive experimental effort, few theoretical justifications or computable formalizations exist in the literature. Sachdeva & McAuley (2023) proposes a high-level formulation based on minimizing the difference in test loss between learning on the full dataset versus the synthetic dataset. Kungurtsev et al. (2024) considers dataset distillation as dependent on the desired inference task (typically classification with cross-entropy loss for image data), interpreting trajectory matching as a mean-field control problem. No prior literature on dataset distillation addresses theoretically whether or not the distilled datasets are reasonable approximations of the input training data distribution. In this work, we address convergence in measure space for disentangled methods, demonstrating consistency and convergence of the distilled datasets.

We summarize the contributions of this work as follows.

1. We theoretically justify the disentangled dataset distillation framework, exploiting its structure to show that these methods converge to the true data distribution as the number of distilled points increases, using classical notions of optimal quantization and Wasserstein distance. Motivated by the empirical usage of clustering in latent spaces, we show in Theorem 1 that optimal quantizations induce convergent approximations of gradients of population risk. Furthermore, Corollary 1 shows the approximation rate is given by $\mathcal{O}(K^{-1/d})$, where $d$ is the dimension of the latent space and $K$ is the number of quantization points. This motivates the usage of a low-dimensional latent space to model the data distribution.

2. We propose *Dataset Distillation by Optimal Quantization* (DDOQ) in Algorithm 1, a DD algorithm based on clustering in a low-dimensional latent space. Compared to a recent disentangled method $D^4M$, our proposed method has a smaller Wasserstein distance between the distilled latent points and latent data distribution, indicating better approximation.

3. We algorithmically compare our proposed method with $D^4M$ and various common disentangled baselines on the ImageNet-1K dataset using the same generative diffusion model backbone, demonstrating significantly better classification accuracy at varied IPC budgets and better cross-architecture generalization. To demonstrate the potential of DDOQ, we additionally use the stronger diffusion transformer (DiT) backbone, used in recent diffusion-based methods. We provide a central comparison with SOTA disentangled and diffusion-based distillation methods, yielding competitive or better results than existing SOTA methods on ImageNet-1K and its subsets.

This work is structured as follows. Section 2 covers related background and convergence results, including optimal quantization and diffusion models. Theorem 1 in Section 3 demonstrates consistency of the optimal quantizers when passed through diffusion-based generative priors to the image space, and motivates adding automatically-learned weights in the prototyping phase. Section 3.1 details the data distillation pipeline in a sequential manner, from distillation to training new models using the computed weights. Section 4 contains experiments of our proposed method against the previously SOTA $D^4M$ method as well as other recent SOTA baselines on the large-scale ImageNet-1K dataset.

## 2 BACKGROUND

Define $\mathcal{P}_2(\mathbb{R}^d)$ to be the set of probability measures on $\mathbb{R}^d$ with finite second moment, not necessarily admitting a density with respect to the Lebesgue measure. We use $\mathcal{W}_2$ to denote the Wasserstein-2 distance between two probability distributions in $\mathcal{P}_2(\mathbb{R}^d)$ (Santambrogio, 2015).

## 2.1 Optimal Quantization

For a probability measure $\mu \in \mathcal{P}_2(\mathbb{R}^d)$, an *optimal quantization* (or vector quantization) at level $K$ is a set of points $\{x_1, ..., x_K\} \subset \mathbb{R}^d$ such that the $\mu$-averaged Euclidean distance to the quantized points is minimal. This can be formulated as the minimizer of the (quadratic) distortion, defined as follows.

**Definition 1** (Quadratic distortion). *For a quantization grid* $(x_1, ..., x_K) \in (\mathbb{R}^d)^K$, *the corresponding* Voronoi cells *are*

$$C_i = \{y \in \mathbb{R}^d \mid \|y - x_i\| = \min_j \|y - x_j\|\}, \quad i = 1, ..., K. \tag{2}$$

*Given a measure* $\mu \in \mathcal{P}_2(\mathbb{R}^d)$, *the* (quadratic) distortion *function* $\mathcal{G} = \mathcal{G}_\mu$ *takes a tuple of points* $(x_1, ..., x_K)$ *and outputs the average squared distance to the set:*

$$\mathcal{G} : (x_1, ..., x_K) \mapsto \int_{\mathbb{R}^d} \min_i \|x - x_i\|^2 \, \mu(\mathrm{d}x) = \mathbb{E}_{X \sim \mu}[\min_i \|X - x_i\|^2]. \tag{3}$$

*We will write* $\mathcal{G}_{K,\mu}$ *to mean the distortion function at level* $K$, *i.e. with domain* $(\mathbb{R}^d)^K$, *and drop the subscripts where it is clear. An* optimal quantization *is a minimizer of* $\mathcal{G}$.

Note that the assumption that $\mu \in \mathcal{P}_2(\mathbb{R}^d)$ has finite second moments implies that the quadratic distortion is finite for any set of points, and that an optimal quantization exists (Pagès, 2015). We note that the optimal quantization weights are uniquely determined by the quantization points. This gives equivalence of the distortion minimization problem to the Wasserstein minimization problem, when restricted to measures of finite support (Pagès, 2015).

**Proposition 1.** *Suppose we have a quantization* $\mathbf{x} = \{x_1, ..., x_K\}$. *Assume that the (probability) measure* $\mu$ *is null on the boundaries of the Voronoi cells* $\mu(\partial C_i) = 0$. *Then the measure* $\nu$ *that minimizes the Wasserstein-2 distance* (14) *and satisfies* $\operatorname{supp} \nu \subset \{x_1, ..., x_K\}$ *is* $\nu_K = \sum_{i=1}^K \mu(C_i)\delta(x_i)$, *where* $\delta(x_i)$ *denotes the Dirac delta distribution at* $x_i$. *Moreover, the optimal coupling is given by the projection onto the centroids.*

*Proof.* Deferred to Section C.2. □

In other words, finding points that minimize the quadratic distortion is equivalent to finding a $K$-finitely supported (probability) measure, minimizing the Wasserstein-2 distance to the underlying measure.

**Remark 1.** *The case where the approximating measure is a uniform Dirac mixture is called the* Wasserstein barycenter *problem (Cuturi & Doucet, 2014). The Wasserstein barycenter has higher error than the optimal quantization, but it admits an easily computable dual representation.*

The quantizer can be shown to have nice approximation properties when taking expectations of functions. A prototypical example for DD would have $f$ be the gradient of a neural network with respect to some loss function. This implies that a data distribution $\mu$ and its quantization $\nu$ induce similar training dynamics.

**Proposition 2.** *Let* $f : \mathbb{R}^d \to \mathbb{R}$ *be an L-Lipschitz function. For a probability measure* $\mu \in \mathcal{P}_2(\mathbb{R}^d)$ *that assigns no mass to hyperplanes, and a quantization* $\mathbf{x} = (x_1, ..., x_k)$, *let* $\nu = \sum_{i=1}^K \mu(C_i)\delta(x_i)$ *be the corresponding Wasserstein-optimal measure with support in* $\mathbf{x}$, *as in Proposition 1. The difference between the population risk* $\mathbb{E}_\mu[f]$ *and the weighted empirical risk* $\mathbb{E}_\nu[f]$ *is bounded as*

$$\mathbb{E}_\mu[f] - \mathbb{E}_\nu[f] \leq L\mathcal{G}(\mathbf{x})^{1/2}. \tag{4}$$

*Proof.* Deferred to Section C.3. □

### 2.1.1 Solving the optimal quantization problem

To find an optimal quantizer, we use the competitive learning vector quantization (CLVQ) algorithm (Ahalt et al., 1990). This arises directly from gradient descent on the quadratic distortion. The gradient of the distortion has a representation in terms of $\mu$-centroids of the corresponding Voronoi cells, given explicitly in Section C.5.

The CLVQ algorithm is presented in the appendix as Algorithm 2. It consists of iterating: (i) sampling from $\mu$, (ii) computing the nearest cluster centroid, and (iii) updating the cluster centroid with weighted average. CLVQ is *equivalent to the mini-batch k-means method* when the step-sizes $\gamma_i$ are chosen to be the reciprocals of the number of points per-cluster (Sculley, 2010; Pedregosa et al., 2011). We can thus interpret mini-batch $k$-means as finding a local minima of the optimal quantization problem.

The CLVQ algorithm produces points $\mathbf{x} = (x_1, ..., x_K)$, but it remains to compute the associated weights approximating the measures of the Voronoi cells $\mu(C_i)$ as in Proposition 1. This can be done in an online manner within the same iterations (Pagès, 2015).

**Proposition 3** (Bally & Pagès 2003, Prop. 7). *Assume that the measure $\mu \in \mathcal{P}_{2+\eta}(\mathbb{R}^d)$ for some $\eta > 0$, and that it assigns no mass to hyperplanes. Assume further that the grids $\mathbf{x}^{(t)}$ produced by CLVQ converge to a stationary grid $\mathbf{x}^*$, i.e. $\nabla \mathcal{G}(\mathbf{x}^*) = 0$, and that the step-sizes satisfy $\sum \gamma_k = +\infty$ and $\sum \gamma_k^{1+\delta} < \infty$ for some $\delta > 0$. Then,*

1. *The companion weights $w_k$ converge almost surely to the limiting weights $\mu(C_k^*)$;*

2. *The moving average of the empirical quadratic distortion converges to the limiting distortion:*

$$\frac{1}{t} \sum_{k=1}^{t} \min_{1 \leq i \leq K} \|X_k - x_i^{(k)}\|^2 \to \mathcal{G}(\mathbf{x}^*).$$

Using these weights, we target the problem of approximating optimal quantizers, rather than the Wasserstein barycenter problem. The addition of the weights reduces the distortion in the latent space. In the following section, we show that this reduced distortion carries over to the image space, which leads to better training fidelity as given using Proposition 2.

## 2.2 SCORE-BASED DIFFUSION

To connect the quantization error on the latent space with the quantization error in the image space, we need to consider properties of the latent-to-image process. In particular, we focus on score-based diffusion models, seen as discretizations of particular noising SDEs (Song et al., 2020). Consider the following SDE, where $W$ is a standard Wiener process:

$$\mathrm{d}x = f(x, t)\, \mathrm{d}t + g(t)\, \mathrm{d}W. \tag{5}$$

The reverse of the diffusion process is given by the reverse-time SDE (Haussmann & Pardoux, 1986), where the density of $x$ at time $t > 0$ is given by $p_t$,

$$\mathrm{d}x = [f(x, t) - g(t)^2 \nabla_x \log p_t(x)]\, \mathrm{d}t + g(t)\mathrm{d}\bar{W}, \tag{6}$$

and $\bar{W}$ is a reverse time Wiener process. For an increasing noising schedule $\sigma(t)$ or noise-scale $\beta(t)$, the variance-exploding SDE (VESDE, or Brownian motion) and variance-preserving SDE (VPSDE, or Ornstein–Uhlenbeck process) are given respectively by

$$\mathrm{d}x = \sqrt{\frac{\mathrm{d}[\sigma^2(t)]}{\mathrm{d}t}}\, \mathrm{d}W \quad \text{and} \quad \mathrm{d}x = -\frac{1}{2}\beta(t)x\, \mathrm{d}t + \sqrt{\beta(t)}\, \mathrm{d}W. \tag{7}$$

These SDEs are related to early diffusion models. Specifically, VPSDE corresponds to denoising score matching with Langevin dynamics (Song & Ermon, 2019), and VESDE to denoising diffusion probabilistic models (Sohl-Dickstein et al., 2015; Ho et al., 2020). Using this particular structure, we may obtain convergence results as seen in the next section. We note that by time-rescaling, we may assume without loss of generality that the noising schedule is linear $\sigma^2(t) = t$ or the noise-scale is constant $\beta(t) = 1$.

The goal in question: given Wasserstein-2 convergence of the marginals $\nu_T^{(k)} \to \mu_T$, we wish to derive a bound on expectations $\mathbb{E}_{\nu_\delta^{(k)}}[f] \xrightarrow{?} \mathbb{E}_{\mu_\delta}[f]$, for some small fixed $\delta \in (0, T)$ and $f : \mathbb{R}^d \to \mathbb{R}$ satisfying some regularity conditions. In other words, we wish to show that generative diffusion preserves closeness of data distributions. Such a bound would directly link to training neural networks with surrogate data, e.g. by taking $f$ to be the gradient of a loss function with respect to some network parameters.

**Remark 2.** *Having $\delta = 0$ may not be well defined because of non-smoothness and blowup of the score at time 0 for singular measures (Pidstrigach, 2022; Yang et al., 2023).*

**Remark 3.** *Working with weak convergence is necessary due to the singular empirical measures. Pidstrigach (2022) demonstrates that the backward SDE process satisfies a data-processing inequality, showing that the $f$-divergence after backwards diffusion is at most the $f$-divergence at marginal time $T$. However, $f$-divergences require absolute continuity of the compared marginal with respect to the underlying diffused distribution, which is equivalent to absolute continuity with respect to the Lebesgue measure by the Hörmander condition. This rules out singular initializations such as empirical measures, which arise in dataset distillation.*

# 3 DATASET DISTILLATION AS OPTIMAL QUANTIZATION

Our main result Theorem 1 gives consistency of dataset distillation for a score-based diffusion prior in the image space. Later, we use this in Section 3.1 to present DDOQ as a modification of the D$^4$M method. By simply changing the clustering objective from a Wasserstein barycenter to an optimal quantization by adding weights, we can effectively reduce the Wasserstein distance to the data distribution. Moreover, from Proposition 3, the weights are automatically determined during the $k$-means clustering process.

**Theorem 1.** *Consider the VESDE/Brownian motion or the VPSDE/Ornstein–Uhlenbeck process*

$$\mathrm{d}x = \mathrm{d}W \quad or \quad \mathrm{d}x = -\frac{1}{2}x\,\mathrm{d}t + \mathrm{d}W\,. \tag{8}$$

*For any initial data distribution $\mu \in \mathcal{P}_2(\mathbb{R}^d)$ with compact support bounded by $R > 0$, the backwards diffusion process is well posed. Suppose that there are two distributions $\mu_T, \nu_T$ at time $T$ that undergo the reverse diffusion process (with fixed initial reference measure $\mu$) up to time $t = \delta \in (0, T)$ to produce distributions $\mu_\delta, \nu_\delta$. There exists a (universal explicit) constant $C = C(\delta, T, R, d) \in (0, +\infty)$ such that if $f : \mathbb{R}^d \to \mathbb{R}^n$ is an L-Lipschitz function, then the difference in expectation satisfies*

$$\|\mathbb{E}_{\mu_\delta}[f] - \mathbb{E}_{\nu_\delta}[f]\| \leq CL\mathcal{W}_2(\mu_T, \nu_T). \tag{9}$$

In dataset distillation terms, $f$ will typically be replaced by the gradient of a loss function. The above result suggests that a distilled *image* dataset can be given by passing a distilled *latent* dataset through the generative reverse SDE process. Moreover, when training a neural network on a distilled dataset given by optimal quantization, the gradients on the distilled dataset and full training dataset at each step will automatically be similar. This bypasses the heuristics needed in bi-level DD formulations, and avoids fine-tuning or generation-time guidance of the diffusion models.

Theorem 1 combined with the asymptotic rates of Graf & Luschgy (2000) gives convergence rates as the number of quantization points increases. We note that this can be further be combined with the convergence of optimal quantization rates of empirical measures such as Theorem 3 in Section C.4. In particular, the next result shows that as the number of points increases, we have convergence to the underlying data distribution in image space, giving *consistency*.

**Corollary 1.** *Suppose $\mu \in \mathcal{P}_2(\mathbb{R}^d)$ has compact support and is diffused through either the Brownian motion or Ornstein–Uhlenbeck process up to time $T$ to produce marginal $\mu_T$. Let $\nu_T^{(K)}$ be optimal quantizers of $\mu_T$ at level $K$ for $K \in \mathbb{N}$. For fixed $\delta \in (0, T)$, let $\nu_\delta^{(K)}$ denote the corresponding backwards diffusion at time $T - \delta$. Then, for any L-Lipschitz function $f$ and as $K \to \infty$,*

$$\|\mathbb{E}_{\mu_\delta}[f] - \mathbb{E}_{\nu_\delta^{(K)}}[f]\| = L\mathcal{O}(K^{-1/d}). \tag{10}$$

## 3.1 PROPOSED METHOD: DATASET DISTILLATION BY OPTIMAL QUANTIZATION

We have seen that clustering gives a consistent approximation to the latent distribution. Using the generative diffusion model and Theorem 1, we obtain a consistent approximation to the original data distribution in the image space. We now propose *Dataset Distillation by Optimal Quantization* (DDOQ). We detail the steps in text below and summarize in Algorithm 1.

Suppose we are given a latent diffusion model (LDM), i.e. a (conditional) encoder-decoder pair and a diffusion model on the latent space (Rombach et al., 2022). Let $K$ be the target number of images per class. Constructing and using the distilled dataset consists of the following steps in sequence.

---

**Algorithm 1:** Dataset Distillation by Optimal Quantization (DDOQ)

---

**Data:** Training data and labels $(\mathcal{T}, \mathcal{L})$, pre-trained encoder-decoder pair $(\mathcal{E}, \mathcal{D})$, text encoder $\tau$, latent diffusion model $\mathcal{U}_t$, target IPC count $K$

```
/* Step 1:   encode                                                            */
```
1  Initialize latent points $Z = \mathcal{E}(\mathcal{T})$;
```
/* Step 2:   cluster                                                           */
```
2  Compute and save $k$-means cluster centers $z_k^{(L)}$ and cluster counts $v_k^{(L)}$, $k = 1, ..., K$, $L \in \mathcal{L}$;
3  Compute weights $w_k^{(L)} \leftarrow v_k^{(L)} / \sum_j v_j^{(L)}$;
```
/* Step 3:   decode                                                            */
```
4  Compute class embeddings $\text{emb} = \tau(L)$, $L \in \mathcal{L}$;
5  Compute and save class distilled images $\mathcal{S}_L = \{x_k^{(L)} = \mathcal{D} \circ \mathcal{U}_t(z_k^{(L)}, \text{emb}) \mid k = 1, ..., K\}$;
   **Result:** Distilled images $\mathcal{S} = \bigcup_{L \in \mathcal{L}} \mathcal{S}_L$
6  **To train a new network $f_\theta$:**
   **Data:** Labels $y$ for training data $x \in \mathcal{S}$, loss function $\ell : (x, y, \theta) \rightarrow \mathbb{R}$
```
/* Step 4:   Train new model (Validation)                                      */
```
7  Train network using loss function $\min_\theta \sum_{(x,y,w)} w \cdot \ell(x, y, \theta)$

---

Table 1: Wasserstein-2 distances of the distilled latents compared to the encoded training data on classes of ImageNet-1K. We observe the weighting drops the Wasserstein-2 distance significantly.

| Test class | | 0 | 1 | 2 | Avg reduction |
|---|---|---|---|---|---|
| IPC 10 | D⁴M | 49.67 | 51.14 | 47.16 | $-15.7\%$ |
| | DDOQ | 42.97 | 42.71 | 39.14 | |
| IPC 50 | D⁴M | 47.34 | 49.10 | 45.28 | $-16.1\%$ |
| | DDOQ | 40.41 | 41.01 | 37.53 | |

1. **Encoding.** We use the encoder of the LDM to map training samples from the image space to the latent space, giving empirical samples from the SDE marginal $\mu_T$. This is done per-class in the context of classification.

2. **Clustering.** We then use the CLVQ (mini-batch $k$-means) algorithm on these samples to compute the $K$ centroids and corresponding weights, as in Proposition 3. This gives an empirical distribution $\nu_T^{(K)}$ that approximates $\mu_T$. Equivalently, it consists of tuples of points and weights $(x_i, w_i)_{i=1}^K$, such that $\nu_T^{(K)} = \sum_i w_i \delta(x_i) \approx \mu_T$.

3. **Decoding/Image synthesis.** Given the clustered centroids in the latent space, the generative part of the LDM is employed to reconstruct images. This comprises the image component of the distilled dataset. The weights of the latent points are assigned directly to the weights of the corresponding generated image.

4. **Training new models.** Training a network $f_\theta$ from scratch requires: distilled images $x$, some corresponding labels $y$, and the weights $w$ to each image. For a loss function $\ell(x, y, \theta)$ such as cross-entropy or KL divergence, the loss for each sample is weighted by $w$. The complete loss function is given by

$$\min_\theta \sum_{(x,y,w)} w \cdot \ell(x, y, \theta).$$

Within Section 1, the encoder and decoder are given by a pre-trained encoder-decoder model, combined with a diffusion model in the latent space. Compared with D⁴M, we include (automatically determined) weights when training a new network in the final step, indicated by the variable length arrows. These are justified by considering the expectation of network gradients with respect to the training and distilled distributions. As seen in Table 1, the inclusion of the weights significantly decreases the Wasserstein distance of the distilled points to the data distribution, when tested on classes of ImageNet-1K.

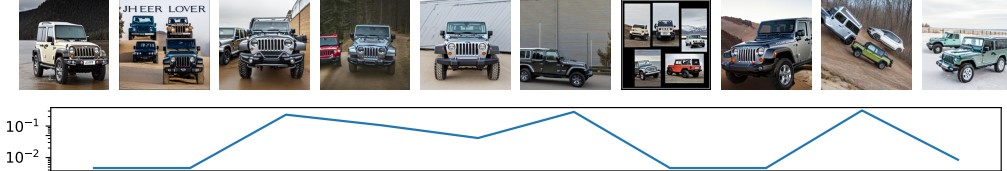

Figure 2: Example distilled images of the "jeep" class in ImageNet-1K along with their $k$-means weights below. There are little to no qualitative features that can be used to differentiate the low and high weighted images, mainly due to the high fidelity of the diffusion model. However, the weights are indicative of the distribution of the training data in the latent space of the diffusion model.

Table 2: Comparison of top-1 classification performance on the ImageNet-1K dataset at various IPCs. We observe that the proposed DDOQ outperforms the clustering-based method $D^4M$, due to the addition of weights to the synthetic data. The maximum performance for all methods should be 69.8 as the soft labels are computed using a pre-trained ResNet-18 model. In particular, we achieve a 30% reduction in error gap using ResNet-101 at IPC 200.

| IPC | Method | ResNet-18 | ResNet-50 | ResNet-101 | IPC | Method | ResNet-18 | ResNet-50 | ResNet-101 | Full |
|---|---|---|---|---|---|---|---|---|---|---|
| 10 | TESLA | 7.7 | - | - | 50 | SRe$^2$L | $46.8_{\pm0.2}$ | $55.6_{\pm0.3}$ | $60.8_{\pm0.5}$ | |
| | SRe$^2$L | $21.3_{\pm0.6}$ | $28.4_{\pm0.1}$ | $30.9_{\pm0.1}$ | | CDA | 53.5 | 61.3 | 61.6 | |
| | RDED | $\mathbf{42.0}_{\pm0.1}$ | - | $\mathbf{48.3}_{\pm1.0}$ | | RDED | $\mathbf{56.5}_{\pm0.1}$ | - | $61.2_{\pm0.4}$ | |
| | D$^4$M | 27.9 | 33.5 | 34.2 | | D$^4$M | 55.2 | 62.4 | 63.4 | 69.8 |
| | DDOQ | $33.1_{\pm0.60}$ | $34.4_{\pm0.99}$ | $36.7_{\pm0.80}$ | | DDOQ | $56.2_{\pm0.07}$ | $\mathbf{62.5}_{\pm0.24}$ | $\mathbf{63.6}_{\pm0.13}$ | |
| 100 | SRe$^2$L | $52.8_{\pm0.3}$ | $61.0_{\pm0.4}$ | $62.8_{\pm0.2}$ | 200 | SRe$^2$L | $57.0_{\pm0.4}$ | $64.6_{\pm0.3}$ | $65.9_{\pm0.3}$ | |
| | CDA | 58.0 | 65.1 | 65.9 | | CDA | 63.3 | 67.6 | 68.4 | |
| | D$^4$M | 59.3 | 65.4 | 66.5 | | D$^4$M | 62.6 | 67.8 | 68.1 | |
| | DDOQ | $\mathbf{60.1}_{\pm0.15}$ | $\mathbf{65.9}_{\pm0.15}$ | $\mathbf{66.7}_{\pm0.06}$ | | DDOQ | $\mathbf{63.4}_{\pm0.08}$ | $\mathbf{68.0}_{\pm0.05}$ | $\mathbf{68.6}_{\pm0.08}$ | |

We note that there is flexibility in the choice of label $y$ when training a new model. For example, the soft-label synthesis of $D^4M$ or RDED employs another pre-trained classifier $\Psi$, such as a ResNet. Using this, the soft-labels are given by $y = \Psi(x)$, as opposed to one-hot encodings of the corresponding classes.

## 4 EXPERIMENTS

We compare with two different latent diffusion architectures, namely the original latent diffusion model utilizing UNets (Rombach et al., 2022), then the stronger diffusion transformer (DiT) architecture (Peebles & Xie, 2023). We first use the LDM to compare the pure algorithmic differences of DDOQ with $D^4M$ and related baselines. Then, we use DiT to demonstrate the potential of DDOQ compared to newer state-of-the-art baselines.

### 4.1 UNET BACKBONE

To validate the proposed DDOQ algorithm, we directly compare with the previous state-of-the-art disentangled methods $D^4M$ (Su et al., 2024) and RDED (Sun et al., 2024) on the ImageNet-1K dataset. Baseline figures are reported as given in their respective works. For low IPC, we also report the TESLA method, which is a SOTA bi-level method based on MTT (Cui et al., 2023), but is unscalable past IPC=10. RDED achieves strong results for low IPC using its aggregated images and special training schedule, while CDA improves upon SRe$^2$L using time-varying augmentation. For consistency, soft labels are computed using the pre-trained PyTorch ResNet-18 model, and new ResNet-{18,50,101} models are trained using the distilled data. We provide a direct comparison of the distilled data performance in Table 2 for the IPCs $K \in \{10, 50, 100, 200\}$. Variances for DDOQ are averaged over five models trained on the same distilled data.

We observe that while RDED is very powerful in the low IPC setting due to the patch-based distilled images, which effectively gives the information of 4 (down-sampled) images in one training sample.

Table 3: Generalization performance of $D^4M$ and the proposed DDOQ method for different soft-label teachers and student architectures at IPC 50. We observe that DDOQ has uniformly better cross-architecture generalization for convolutional teacher and student architectures, and slightly worse performance for student models using the transformer architecture Swin-T.

| Teacher Network | | Student Network | | | |
|---|---|---|---|---|---|
| | | ResNet-18 | MobileNet-V2 | EfficientNet-B0 | Swin-T |
| ResNet-18 | $D^4M$ | 55.2 | 47.9 | 55.4 | **58.1** |
| | DDOQ | **56.2** | **52.1** | **58.0** | 57.4 |
| MobileNet-V2 | $D^4M$ | 47.6 | 42.9 | 49.8 | **58.9** |
| | DDOQ | **47.7** | **45.6** | **52.5** | 56.3 |
| Swin-T | $D^4M$ | 27.5 | 21.9 | 26.4 | **38.1** |
| | DDOQ | **28.5** | **24.1** | **29.3** | 36.0 |

However, the gap quickly reduces for IPC 50, getting outperformed by the clustering-based $D^4M$ and proposed DDOQ methods with more powerful models like ResNet-101. Results for IPC 100/200 are not available online for RDED and we omit them due to computational restriction.

The proposed DDOQ algorithm is uniformly better than $D^4M$, with the most significant increase in the low IPC setting. Moreover, DDOQ surpasses all the compared SOTA disentangled DD methods for IPC 100 and 200. For a low number of quantization points, the gap in Wasserstein distance of the Wasserstein barycenter and the optimal quantizer to the data distribution may be large. As indicated in Theorem 1, a lower Wasserstein distance means more faithful gradient computations on the synthetic data, which may explain the higher performance with minimal algorithmic change, as well as implicitly allowing for gradient matching as in existing bi-level methods.

Table 3 shows the generalization performance of the distilled latent points. The PyTorch pre-trained MobileNet-V2 and Swin-T networks are used to create the soft labels. We then evaluate the distilled images and soft labels with three convolutional architectures and the Swin-T transformer architecture. We observe that DDOQ is not only able to generalize to different model architectures, but also uniformly outperforms $D^4M$ on all convolutional student architectures. The slightly worse performance of DDOQ when using a transformer architecture for the student may be due to more precise hyperparameter tuning requirements.

To illustrate the weights, Figure 2 plots ten example images from the "jeep" class when distilled using $K = 10$ IPC. We observe that there is a very large variance in the weights $v_k^{(L)} / \sum_{j=1}^K v_j^{(L)}$, indicating the presence of strong clustering in the latent space. Nonetheless, there is no qualitative evidence that the weights indicate "better or worse" training examples, rather indicating the structure in the latent space.

## 4.2 DiT BACKBONE

To compare the potential of DDOQ, we use a stronger generative model, namely the diffusion transformer (DiT) (Peebles & Xie, 2023). This architecture achieves uniformly better sample quality compared to the LDM used in the previous section, in terms of Inception Score and Fréchet inception distance. We denote the method with DiT backbone as DDOQ-DiT.

We compare with the SOTA DD methods based on diffusion guidance, namely Minimax (Gu et al., 2024) and Influence Guided Diffusion (IGD) (Chen et al., 2025), which both use DiT. These methods guide the decoding process using some fine-tuning or batch statistics. In contrast, DDOQ-DiT directly modifies the initialization in the latent space. In addition, we compare with samples directly generated with DiT from random initializations, labelled 'DiT'. We compare on ImageNet-1K, as well as the 10-class subsets ImageNette and ImageWoof. The baseline figures are taken from Chen et al. (2025) which report higher numbers than Gu et al. (2024).

Table 4 demonstrates that we significantly outperform the diffusion-guidance based methods on the full ImageNet-1K, as well as DDOQ-DiT with the UNet backbone. Moreover, we are competitive-

Table 4: Comparisons on ImageWoof and ImageNette using the ResNetAP-10 architecture, and ImageNet-1K using ResNet-18. DDOQ-DiT outperforms the guided diffusion methods at low IPC and is competitive at higher IPC, namely outperforming the compared methods on ImageNet-1K.

| Dataset | IPC | DiT | DiT-IGD | Minimax | Minimax-IGD | DDOQ-DiT | Full |
|---|---|---|---|---|---|---|---|
| ImageWoof | 10 | $39.0_{\pm0.9}$ | $41.0_{\pm0.8}$ | $39.6_{\pm1.2}$ | $43.3_{\pm0.3}$ | $\mathbf{48.8}_{\pm2.0}$ | 87.2 |
| | 50 | $55.8_{\pm1.1}$ | $62.7_{\pm1.2}$ | $59.8_{\pm0.8}$ | $65.0_{\pm0.8}$ | $\mathbf{65.4}_{\pm0.7}$ | |
| ImageNette | 10 | $62.8_{\pm0.8}$ | $66.5_{\pm1.1}$ | $63.2_{\pm1.0}$ | $65.3_{\pm1.1}$ | $\mathbf{68.2}_{\pm0.9}$ | 94.6 |
| | 50 | $76.9_{\pm0.5}$ | $81.0_{\pm1.2}$ | $78.2_{\pm0.7}$ | $\mathbf{82.3}_{\pm1.1}$ | $79.8_{\pm0.8}$ | |
| ImageNet-1K | 10 | $39.6_{\pm0.4}$ | $45.5_{\pm0.5}$ | $44.3_{\pm0.5}$ | $46.2_{\pm0.6}$ | $\mathbf{53.0}_{\pm0.2}$ | 69.8 |
| | 50 | $52.9_{\pm0.6}$ | $59.8_{\pm0.3}$ | $58.6_{\pm0.3}$ | $60.3_{\pm0.4}$ | $\mathbf{62.7}_{\pm0.1}$ | |

with or better than the baselines on the 10-class subsets, namely outperforming in the low IPC setting and in the more difficult ImageWoof subset. Moreover, the stronger diffusion model significantly benefits the low IPC ImageNet-1K setting, increasing ResNet-18 test accuracy from 33.1 to 53.0. An ablation on the effect of different weights during training is given in Section I.

## 5 CONCLUSION

This work proposes DDOQ, a dataset distillation method based on optimal quantization. Inspired by optimal quantization and Wasserstein theory, we theoretically demonstrate consistency of the distilled datasets in Theorem 1 when using standard diffusion-based generative models to generate the synthetic data. Experiments show the proposed method is competitive or better than SOTA on large-scale ImageNet experiments.

We have presented theoretical justification for disentangled dataset distillation methods, which rely on clustering as the mechanism for approximating the underlying data distributions. More specifically, we justify the combination of a low-dimensional latent space and a diffusion model. Consistency or convergence of other dataset distillation frameworks such as bi-level methods and diffusion guidance are still open questions.

Future work could include sharper bounds in Theorem 1 that exploit the sub-Gaussianity of the diffused distributions or manifold hypothesis. Other interesting directions could be relating the weightings of the synthetic data to the hardness of learning the data, such as in (Joshi & Mirzasoleiman, 2023), further increasing the performance using different training regimes such as curriculum learning, or extending the theoretical analysis to inexact score matching. Possible empirical work could investigate correlations between the weights and influence (Pruthi et al., 2020).

### ACKNOWLEDGMENTS

This work was done while HYT was on placement in GSK.ai. HYT acknowledges support from GSK and the Masason Foundation.

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

## A    DETAILS ON HEURISTIC BI-LEVEL DISTILLATION

1. (*Meta-learning.*) This uses the assumption that more data produces better results, replacing the risk matching objective with a risk minimization objective and removing the dependence on $\mathcal{T}$ (Wang et al., 2018; Deng & Russakovsky, 2022). The learning algorithm $\Phi_S = \Phi_S(\theta_0)$ with initial parameter $\theta_0$ is given by unrolling gradient steps $\theta_{t+1} = \theta_t - \eta \nabla \mathcal{L}_S(\theta_t)$. The distilled dataset minimizes the training loss as trained on the distilled set up to $T$ iterations:

$$\arg \min_S \mathbb{E}_{\theta_0 \sim p_\theta} \left[ \mathcal{L}_\mathcal{T}(\theta_T) \right]. \tag{11}$$

2. (*Distribution matching.*) Inspired by reproducing kernel Hilbert spaces and the empirical approximations using random neural network features, Zhao & Bilen (2023) proposes an alternate distillation objective that is independent of target loss. The minimization objective is

$$\arg \min_S \mathbb{E}_{\theta \sim p_\theta} \| \frac{1}{|\mathcal{T}|} \sum_{x \in \mathcal{T}} \psi_\theta(x) - \frac{1}{|\mathcal{S}|} \sum_{x \in \mathcal{S}} \psi_\theta(x) \|^2, \tag{12}$$

where $\psi_\theta$ are randomly initialized neural networks. This can be intuitively interpreted as matching the (first moment of) neural network features over the synthetic data.

3. (*Trajectory matching.*) For a *fixed network architecture*, this method aims to match the gradient information of the synthetic and true training datasets from different initializations (Cazenavette et al., 2022). The heuristic is that similar network parameters will give similar performance. In addition, the gradients are allowed to be accelerated by matching a small number of synthetic training steps with a large number of real data training steps: for $N \ll M$ steps, the Matching Training Trajectory (MTT) objective is (with abuse of notation):

$$\arg \min_S \mathbb{E}_{\theta_0 \sim p_\theta} \sum_{t=1}^{T-M} \frac{\|\theta_{t+N}^\mathcal{S} - \theta_{t+M}^\mathcal{T}\|^2}{\|\theta_{t+M}^\mathcal{T} - \theta_t^\mathcal{T}\|^2}, \tag{13}$$

where $\theta_{t+1}^\mathcal{T} = \theta_t^\mathcal{T} - \eta \nabla \mathcal{L}_\mathcal{T}(\theta_t^\mathcal{T})$, $\theta_{t+i+1}^\mathcal{S} = \theta_{t+i}^\mathcal{S} - \eta \nabla \mathcal{L}_\mathcal{S}(\theta_{t+i}^\mathcal{S})$, $\theta_{t+0}^\mathcal{S} = \theta_t^\mathcal{T}$.

Practical computational approximations include pre-training a large number of teacher networks from different initializations.

## B    ADDITIONAL DEFINITIONS

The Wasserstein-2 distance is defined as

$$\mathcal{W}_2(\mu, \nu) = \left( \inf_{\gamma \in \Gamma(\mu, \nu)} \iint_{\mathbb{R}^d \times \mathbb{R}^d} \|x - y\|^2 \, \mathrm{d}\gamma(x, y) \right)^{1/2}, \tag{14}$$

where $\Gamma(\mu, \nu)$ denotes the set of all *couplings*, i.e. joint probability measures on $\mathbb{R}^d \times \mathbb{R}^d$ with marginals $\mu$ and $\nu$.

## C    OPTIMAL QUANTIZATION

### C.1    ADDITIONAL BACKGROUND

Optimal quantizers converge in Wasserstein distance at a rate $\Theta(K^{-1/d})$ to their respective distributions as the number of particles $K$ increases (Graf & Luschgy, 2000). Moreover, any limit point of the optimal quantizers is an optimal quantizer of the limiting distribution (Pollard, 1982). Moreover, it can be shown that if a sequence of distributions converges in Wasserstein distance $\mu_n \to \mu$, then so do the errors in quantization for a fixed quantization level (Liu & Pagès, 2020, Thm. 4).

### C.2    PROOF OF PROPOSITION 1

**Proposition 4.** *Suppose we have a quantization* $\mathbf{x} = \{x_1, ..., x_K\}$. *Assume that the (probability) measure* $\mu$ *is null on the boundaries of the Voronoi clusters* $\mu(\partial(C_i)) = 0$. *Then the measure* $\nu$ *that minimizes the Wasserstein-2 distance* (14) *and satisfies* $\mathrm{supp}\, \nu \subset \{x_1, ..., x_K\}$ *is* $\nu_K = \sum_{i=1}^K \mu(C_i)\delta(x_i)$. *Moreover, the optimal coupling is given by the projection onto the centroids.*

*Proof.* For any coupling $\gamma \in \Gamma(\nu, \mu)$ between $\nu$ and $\mu$, we certainly have that

$$\iint \|x - y\|^2 \, \mathrm{d}\gamma(x, y) \geq \iint \mathrm{dist}(\mathbf{x}, y)^2 \, \mathrm{d}\gamma(x, y) = \int \mathrm{dist}(\mathbf{x}, y)^2 \mathrm{d}\mu(y).$$

where the first inequality comes from definition of distance to a set ($\gamma$-a.s.) and the support condition on $\nu$, and the equality from the marginal property of couplings. The final term is attained when $\nu$ has the prescribed form: the coupling is $c(x_i, y) = \mathbf{1}_{y \in C_i}$ and the corresponding transport map is projection onto the set $\mathbf{x}$ (defined $\mu$-a.s. from the null condition). This shows a lower bound of (14) that is attained. $\qquad\square$

## C.3 PROOF OF PROPOSITION 2

**Theorem 2.** *Let $f : \mathbb{R}^d \to \mathbb{R}$ be an L-Lipschitz function. For a probability measure $\mu \in \mathcal{P}_2(\mathbb{R}^d)$ that assigns no mass to hyperplanes, and a quantization $\mathbf{x} = (x_1, ..., x_k)$, let $\nu = \sum_{i=1}^{K} \mu(C_i) \delta(x_i)$ be the corresponding Wasserstein-optimal measure with support in $\mathbf{x}$, as in Proposition 1. The difference between the population risk $\mathbb{E}_\mu[f]$ and the weighted empirical risk $\mathbb{E}_\nu[f]$ is bounded as*

$$|\mathbb{E}_\mu[f] - \mathbb{E}_\nu[f]| \leq L\mathcal{G}(\mathbf{x})^{1/2}. \tag{15}$$

*Proof.* Since $\mu$ assigns no mass to hyperplanes, we may decompose into Voronoi cells.

$$\left| \int_{\mathbb{R}^d} f \, \mathrm{d}\mu - \int_{\mathbb{R}^d} f \, \mathrm{d}\nu \right| = \left| \sum_{i=1}^{K} \int_{C_i} f(x) - f(x_i) \, \mathrm{d}\mu(x) \right|$$

$$\leq \sum_{i=1}^{K} \int_{C_i} L\|x - x_i\| \, \mathrm{d}\mu(x)$$

$$= L \int_{\mathbb{R}^d} \min_i \|x - x_i\| \, \mathrm{d}\mu(x)$$

$$\leq L \left( \int_{\mathbb{R}^d} \min_i \|x - x_i\|^2 \, \mathrm{d}\mu(x) \right)^{1/2}$$

$$= L\mathcal{G}(\mathbf{x})^{1/2}.$$

The first equality comes from definition of $\nu$, and the inequalities from the Lipschitz condition and Hölder's inequality respectively. $\qquad\square$

For a given quantization $\mathbf{x}$ or quantization level $K$, the quadratic distortion thus satisfies respectively:

$$\mathcal{G}(\mathbf{x})^{1/2} = \inf \{\mathcal{W}_2(\nu, \mu) \mid \text{probability measures } \nu, \, \mathrm{supp}\, \nu \subset \mathbf{x}\}; \tag{16}$$

$$\underset{\mathbf{x}, \, |\mathbf{x}|=K}{\arg\min} \mathcal{G}(\mathbf{x}) = \arg\min \{\mathcal{W}_2(\nu, \mu) \mid \text{probability measures } \nu, \, |\mathrm{supp}\, \nu| \leq K\}. \tag{17}$$

## C.4 CONVERGENCE OF OPTIMAL QUANTIZATION

The following result gives a convergence rate, assuming slightly higher regularity of the underlying probability measure.

**Proposition 5** (Liu & Pagès (2020)). *Let $\eta > 0$, and suppose $\mu \in \mathcal{P}_{2+\eta}(\mathbb{R}^d)$. There exists a universal constant $C_{d,\eta} \in (0, +\infty)$ such that for every quantization level,*

$$e_{K,\mu}^* \leq C_{d,\eta} \cdot \sigma_{2+\eta}(\mu) K^{-1/d}, \tag{18}$$

*where $\sigma_r(\mu) = \min_{a \in \mathbb{R}^d} \mathbb{E}_\mu[\|x - a\|^r]^{1/r}$.*

The following non-asymptotic result considers the convergence of optimal quantizers for a sequence of probability measures, converging in the Wasserstein sense.

---

**Algorithm 2:** CLVQ

---

**Data:** initial cluster centers $x_1^{(0)}, ..., x_K^{(0)}$, step-sizes $(\gamma_i)_{i \geq 0}$, $i \leftarrow 0$

1 Initialize weights $\mathbf{w} = (w_1, ..., w_K) = (1/K, ..., 1/K)$;

2 **while** *not converged* **do**

3      Sample $X_i \sim \mu$;

4      Select "winner" $k_{\text{win}} \in \arg\min_{1 \leq k \leq K} \|X_i - x_k^{(i)}\|$;

5      Update $x_k^{(i+1)} \leftarrow (1 - \gamma_i)x_k^{(i)} + \gamma_i X_i$ if $k = k_{\text{win}}$, otherwise $x_k^{(i+1)} \leftarrow x_k^{(i)}$ ;

6      Update weights $w_k \leftarrow (1 - \gamma_i)w_k + \gamma_i \mathbf{1}_{k=k_{\text{win}}}$;

7      $i \leftarrow i + 1$;

8 **end**

**Result:** quantization $\nu_K = \sum_{k=1}^{K} w_i \delta(x_k^*)$

---

**Theorem 3** (Liu & Pagès 2020, Thm. 4). *Fix a quantization level $K \geq 1$. Let $\mu_n, \mu \in \mathcal{P}_2(\mathbb{R}^d)$ with support having at least $K$ points, such that $\mathcal{W}_2(\mu_n, \mu) \to 0$ as $n \to \infty$. For each $n \in \mathbb{N}$, let $\mathbf{x}^{(n)}$ be an optimal quantizer of $\mu_n$. Then*

$$\mathcal{G}_{K,\mu}(\mathbf{x}^{(n)}) - \inf_{\mathbf{x}} \mathcal{G}_{K,\mu}(\mathbf{x}) \leq 4e_{K,\mu}^* \mathcal{W}_2(\mu_n, \mu) + 4\mathcal{W}_2^2(\mu_n, \mu), \tag{19}$$

*where $e_{K,\mu}^* = [\inf_{\mathbf{x}} \mathcal{G}_{K,\mu}(\mathbf{x})]^{1/2}$ is the optimal error.*

**Remark 4.** *Convergence to an optimal quantizer is only guaranteed in the case of a log-concave distribution in one dimension (Kieffer, 1982; Liu & Pagès, 2020). In higher dimensions, convergence to a stationary (but not optimal) grid is possible (Pages & Yu, 2016; Pagès, 2015).*

### C.5 GRADIENT INTERPRETATION OF CLVQ

**Proposition 6** (Differentiability of distortion Pagès 2015, Prop. 3.1). *Let $\mathbf{x} = (x_1, ..., x_K) \in (\mathbb{R}^d)^K$ be such that the $x_i$ are pairwise distinct, and assume that $\mu(\partial C_i) = 0$. Then, the quadratic distortion is differentiable with derivative*

$$\nabla \mathcal{G}(\mathbf{x}) = \left( 2 \int_{C_i(x)} (x_i - \xi) \, \mu(\mathrm{d}\xi) \right)_{i=1,...,K}, \tag{20}$$

*i.e., the gradient for quantization point $x_i$ points away from the $\mu$-centroid of its Voronoi cell.*

The gradient step for some step-sizes $\gamma_k \in (0, 1)$ reads

$$\mathbf{x}^{(k+1)} = \mathbf{x}^{(k)} - \gamma_k \nabla \mathcal{G}(\mathbf{x}^{(k)}), \quad \mathbf{x}^{(0)} \in \text{Hull}(\text{supp}\,\mu)^K, \tag{21}$$

where Hull denotes the convex hull. Recalling that the gradient (20) is a $\mu$-expectation over $C_i(x)$, the corresponding stochastic Monte Carlo version of the above gradient descent yields the CLVQ algorithm 2:

$$\mathbf{x}^{(k+1)} = \mathbf{x}^{(k)} - \gamma_k \left( \mathbf{1}_{X_k \in C_i^{(k)}} x_i^{(k)} - X_k \right)_{1 \leq i \leq K}, \quad X_k \sim \mu. \tag{22}$$

Note that the computation of this requires the ability to sample from $\mu$, as well as being able to compute the nearest neighbor of $X_k$ to the quantization set $\mathbf{x}$ (equivalent to the inclusion $X_k \in C_i^{(K)}$).

### C.6 LLOYD I ALGORITHM

**Lloyd I.** This consists of iteratively updating the centroids with the $\mu$-centroids, given by the $\mu$-average of the Voronoi cells. Clearly, if this algorithm converges, then the centroids are equal to the $\mu$-centroids and the grid is stationary. This is more commonly known as the $k$-means clustering algorithm, employed in common numerical software packages such as `scikit-learn` (Pedregosa et al., 2011). Convergence of the Lloyd-I algorithm can be found in e.g. (Pages & Yu, 2016).

---

**Algorithm 3:** Lloyd I ($k$-means)

---

**Data:** Probability distribution $\mu$ with finite first moment, initial cluster centers $x_1^{(0)}, ..., x_K^{(0)}$

1   $k \leftarrow 0$;

2   **while** *not converged* **do**

3      Compute Voronoi cells $C_i^{(k)} = \{y \in \mathbb{R}^d \mid \|x_i^{(k)} - y\| = \min_j \|x_j^{(k)} - y\|\}$;

4      Replace cluster centers with $\mu$-centroids $x_i^{(k+1)} \leftarrow (\int_{C_i^{(k)}} x\mu(\mathrm{d}x))/\mu(C_i^{(k)})$;

5      $k \leftarrow k + 1$;

6   **end**

---

## D   BACKGROUND ON WELL-POSED DIFFUSIONS

Suppose that the true data distribution on the image space is given by $\mu \in \mathcal{P}(\mathbb{R}^d)$, assumed to have bounded support. Then, by the Hörmander condition, the law of a random variable $(X_t)_{t \geq 0}$ evolving under either the VPSDE or the VESDE will admit a density $p_t(x)$ for all $t > 0$ with respect to the Lebesgue measure, that is smooth with respect to both $x$ and $t$ (Hörmander, 1967). Using the following proposition, we have well-definedness of the backward SDE (Anderson, 1982; Haussmann & Pardoux, 1986).

**Proposition 7.** *For the forward SDE* (5)*, assume that there exists some $K > 0$ such that*

1. *$f(x, t), g(t)$ are measurable, and $f$ is uniformly Lipschitz: $\|f(x,t) - f(y,t)\| \leq K\|x - y\|$ for all $x, y \in \mathbb{R}^d$;*

2. *$\|f(x,t)\| + |g(t)| \leq K(1 + \|x\|)$;*

3. *The solution $X_t$ of* (5) *has a $\mathcal{C}^1$ density $p_t(x)$ for all $t > 0$, and*

$$\int_{t_0}^T \int_{\|x\| < R} \|p_t\|^2 + \|\nabla_x p_t(x)\|^2 < +\infty \quad \forall t_0 \in (0, T], \ R > 0;$$

4. *The score $\nabla \log p_t(x)$ is locally Lipschitz on $(0, T] \times \mathbb{R}^d$.*

*Then the reverse process $X_{T-t}$ is a solution of the* (6)*, and moreover, the solutions of* (6) *are unique in law.*

Now given that the backwards SDE is indeed a diffusion, the data processing inequality uses the Markov property and states that the divergence after diffusion is less than the divergence before diffusion (Liese & Vajda, 2006). This is summarized in (Pidstrigach, 2022, Thm. 1).

**Theorem 4.** *Denote the initial data distribution by $\mu_0 = \mu$, and let $\mu_T$ be the distribution of a random variable $X_t$ satisfying the forward SDE* (5) *on $[0, T]$. Assume the above assumptions, and let $Y_t$ satisfy the backwards SDE* (6) *on $[0, T]$ with terminal condition $Y_T \sim \nu_T$. Denote by $\mu_t$ and $\nu_t$ the marginal distributions at time $t \in [0, T]$ of $X_t$ and $Y_t$, and assume that $\nu_T \ll \mu_T$. Then:*

1. *The limit $Y_0 := \lim_{t \to 0^+} Y_t$ exists a.s., with distribution $\nu_0 \ll \mu_0$.*

2. *For any $f$-divergence $D_f$,*

$$D_f(\mu_0, \nu_0) \leq D_f(\mu_T, \nu_T) \quad and \quad D_f(\nu_0, \mu_0) \leq D_f(\nu_T, \mu_T). \tag{23}$$

This theorem shows that for an $f$-divergence, such as total variation distance or Kullback–Leibler divergence, convergence of the marginals at time $T$ implies convergence of the backwards-diffused marginals at time 0 (also at any $t < T$). However, this requires absolute continuity of the initial marginal distribution $\nu_T$ with respect to $\mu_T$, equivalently, w.r.t. Lebesgue measure. This precludes useful bounds for singular initializations of $\nu_T$, such as empirical measures.

# E  D⁴M ALGORITHM

---

**Algorithm 4:** D⁴M (Su et al., 2024)

---

**Data:** Training data and labels $(\mathcal{T}, \mathcal{L})$, pre-trained encoder-decoder pair $(\mathcal{E}, \mathcal{D})$, text encoder $\tau$, latent diffusion model $\mathcal{U}_t$, target IPC count $K$

1   Initialize latent variables $Z = \mathcal{E}(\mathcal{T})$;

2   **for** *label* $L \in \mathcal{L}$ **do**

3      Initialize latent centroids $z_L^k$, $k = 1, ..., K$;

4      Initialize update counts $v_L^k = 1$, $k = 1, ..., K$;

      /* Compute prototypes with $k$-means                 */

5      **for** *minibatch* $\mathbf{z} \in Z|_L$ **do**

6         **for** $z \in \mathbf{z}$ **do**

7            $\hat{k} \leftarrow \arg\min_k \|z_L^k - z\|$;

             /* Update learning rate                    */

8            $v_L^{\hat{k}} \leftarrow v_L^{\hat{k}} + 1$;

             /* Update centroid                        */

9            $z_L^{\hat{k}} \leftarrow (1 - 1/v_L^{\hat{k}})z_L^{\hat{k}} + (1/v_L^{\hat{k}})z$;

10        **end**

11      **end**

12      Compute class embedding $y = \tau(L)$;

13      Compute class distilled images $\mathcal{S}_L = \{\mathcal{D} \circ \mathcal{U}_t(z_L^k, y) \mid k = 1, ..., K\}$;

14 **end**

**Result:** distilled images $\mathcal{S} = \bigcup_{L \in \mathcal{L}} \mathcal{S}_L$

---

# F  PROOF OF MAIN THEOREM 1

We first require a lemma that controls diffusions for compact measures. In particular, the score is (weakly) monotonic.

**Lemma 1** (Bardet et al. (2018)). *Let $\mu \in \mathcal{P}(\mathbb{R}^d)$ be a probability measure with compact support, say bounded by $R$. Let $g_t(x) = \frac{1}{(2\pi t)^{-d/2}} \exp(-\|x\|^2/2t)$ be the density of the standard normal distribution in $\mathbb{R}^d$. Then if $p_t$ is the density of $\mu * g_t$, it satisfies*

$$\langle x - y, \nabla \log p_t(x) - \nabla \log p_t(y) \rangle \leq \left( \frac{R^2}{t^2} - \frac{1}{t} \right) \|x - y\|^2.$$

*Proof.* From (Bardet et al., 2018, Sec. 2.1), the density $p_t$ can be written as

$$p_t(x) = \frac{1}{(2\pi t)^{d/2}} \exp\left( -\left( \frac{\|x\|^2}{2t} + W_t(x) \right) \right),$$

where

$$W_t(x) = -\log \int_{\mathbb{R}^d} \exp\left( \frac{\langle x, z \rangle}{t} \right) \nu(\mathrm{d}z) - \log C_\nu,$$

with $C_\nu(x) = \int_{\mathbb{R}^d} \exp(-\|x\|^2/2t)\mu(\mathrm{d}x)$ and $\nu(\mathrm{d}x) = C_\nu^{-1} \exp(-\|x\|^2/2t)\mu(\mathrm{d}x)$. Moreover,

$$0 \leq -\nabla^2 W_t \leq \frac{R^2}{t^2} I_d. \tag{24}$$

Therefore,

$$\log p_t(x) = -\frac{d}{2} \log(2\pi t) - \frac{\|x\|^2}{2t} - W_t(x)$$

has Hessian satisfying

$$\nabla^2 \log p_t(x) \leq \left( \frac{R^2}{t^2} - \frac{1}{t} \right) I_d.$$

Therefore, $\nabla \log p_t$ satisfies the desired monotonicity condition. Note that $R$ can be strengthened to $\frac{1}{2}\operatorname{diam}(\operatorname{supp}\mu)$. $\qquad\square$

We next require the following proposition, which can be thought of as a stochastic version of Gronwall's inequality. We present a simple version of the even stronger result available in Hudde et al. (2021).

**Proposition 8** (Hudde et al. 2021, Lem. 3.6). *Consider the diffusion*

$$\mathrm{d}x = f(x, t)\,\mathrm{d}t + g(x, t)\,\mathrm{d}W\,. \tag{25}$$

*Suppose that there exists a measurable $\phi : [0, T] \to [0, \infty]$ satisfying $\int_0^T \phi(t)\,\mathrm{d}t < +\infty$, and that for all $t \in [0, T]$ and $x, y \in \mathbb{R}^d$,*

$$\langle x - y, f(x, t) - f(y, t)\rangle + \frac{1}{2}\|g(x, t) - g(y, t)\|_{\mathrm{HS}}^2 \leq \phi(t)\|x - y\|^2. \tag{26}$$

*Then for processes $X_t^x, X_t^y$ starting at $x, y$ respectively under (25), it holds for all $t \in (0, T]$ that*

$$\|X_t^x - X_t^y\|_{L^1(\Omega;\mathbb{R}^d)} \leq \|x - y\|\exp\left(\int_0^t \phi(s)\,\mathrm{d}s\right). \tag{27}$$

Our goal is to "pushforward" the convergence of the Wasserstein distance from the latent space (of optimal quantizations) through the diffusion model (backwards SDE (6)) into the image space/manifold. We proceed with the proof.

**Theorem 5.** *Consider the VESDE/Brownian motion*

$$\mathrm{d}x = \mathrm{d}W \tag{28}$$

*or the VPSDE/Ornstein–Uhlenbeck process*

$$\mathrm{d}x = -\frac{1}{2}x\,\mathrm{d}t + \mathrm{d}W\,. \tag{29}$$

*Then, for any initial data distribution $\mu \in \mathcal{P}_2(\mathbb{R}^d)$ with compact support bounded by $R > 0$, the assumptions for Proposition 7 hold and the backwards diffusion process is well posed.*

*Suppose further that there are two distributions $\mu_T, \nu_T$ at time $T$ that undergo the reverse diffusion process (with fixed initial reference measure $\mu$) up to time $t = \delta \in (0, T)$ to produce distributions $\mu_\delta, \nu_\delta$. Then there exists a constant $C = C(\delta, R, d) \in (0, +\infty)$ such that if $f : \mathbb{R}^d \to \mathbb{R}^n$ is an $L$-Lipschitz function, then the difference in expectation satisfies*

$$\|\mathbb{E}_{\mu_\delta}[f] - \mathbb{E}_{\nu_\delta}[f]\| \leq CL\mathcal{W}_2(\mu_T, \nu_T). \tag{30}$$

*Proof.* The main idea is to push a Wasserstein-optimal coupling through the backwards SDE, then using Proposition 8 and Proposition 2. First fix a $\delta \in (0, T]$, and let $\mathrm{supp}\,\mu$ be bounded by $R$. Let $g_t = \frac{1}{(2\pi t)^{-d/2}}\exp\left(-\|x\|^2/2t\right)$ denote the distribution of the Gaussian $\mathcal{N}(0, tI_d)$ in $d$ dimensions. By the Hörmander condition, the densities of a random variable $X_t$ with initial distribution $X_0 \sim \mu$ undergoing the Brownian motion or Ornstein–Uhlenbeck process exist, and furthermore the forward and backward SDE processes are diffusions (Malliavin, 1978; Hairer, 2011; Pidstrigach, 2022).

**Step 1.** (*Monotonicity of the drifts.*) For the Brownian motion,

$$p_t(x) = (\mu(z) * g_t)(x).$$

For the Ornstein–Uhlenbeck process, the solution from initial condition $X_0 = x_0$ is

$$x_t = x_0 e^{-t/2} + W_{1-e^{-t}}.$$

The law of $X_t$ is thus $\mu(e^{t/2}x) * g_{1-\exp(-t)}$, where $\mu(e^{t/2}x)$ has support bounded by $Re^{-t/2}$.

Apply Lemma 1 with $\mu$ for the Brownian motion, and $\mu(e^{t/2}x)$ for the Ornstein–Uhlenbeck process. The corresponding backward SDEs (in forward time) for the Brownian motion and Ornstein–Uhlenbeck process as given by the forward time versions of (6) are

$$\mathrm{d}x = \nabla\log p_{T-t}(x)\mathrm{d}t + \mathrm{d}W \quad \text{for Brownian motion;}$$

$$\mathrm{d}x = \left[\frac{x}{2} + \nabla\log p_{T-t}(x)\right]\mathrm{d}t + \mathrm{d}W \quad \text{for the OU process,}$$

where $p_{T-t}(x)$ is the law of $X_{T-t}$ for $t \in [0, T)$.

**Step 2.** (*Lipschitz w.r.t. initial condition after diffusion.*) For the backwards SDEs, since the score is (weakly) monotone and the diffusion term is constant, the backwards SDEs satisfy the monotonicity condition in Proposition 8. Hence, there exists a constant $C$ such that for any $Y_t^x, Y_t^y$ evolving according to the backwards SDE,

$$\mathbb{E}\|Y_{T-\delta}^x - Y_{T-\delta}^y\| \leq C\|x - y\|. \tag{31}$$

From the monotonicity condition and Proposition 8, the constants can be chosen to be as follows, noting the diffusion term is constant:

$$\log C_{\text{Brownian}} = \int_\delta^T \left[\frac{R^2}{t^2} - \frac{1}{t}\right] dt, \quad \log C_{\text{OU}} = \int_\delta^T \left[\frac{R^2 e^{-t}}{(1 - e^{-t})^2} - \frac{1}{1 - e^{-t}}\right] dt. \tag{32}$$

**Step 3.** (*Lift to function expectation.*) Now let $Y_t, \hat{Y}_t$ be two diffusions, initialized with distributions $Y_0^x \sim \mu_T$, $\hat{Y}_t^x \sim \nu_T$. Let $\gamma \in \Gamma(\mu_T, \nu_T)$ be any coupling. Define the "lifted coupling" $\hat{\gamma}$ on the measurable space $\left((\mathbb{R}^d \times \mathbb{R}^d) \times \Omega, \mathcal{B}(\mathbb{R}^d \times \mathbb{R}^d) \otimes \mathcal{F}\right)$, where $(\Omega, \mathcal{F}, \mathbb{P})$ is the underlying (filtered) probability space of the diffusion, as the pushforward of the diffusion:

$$\hat{\gamma} = (Y_0 \mapsto Y_{T-\delta}, \hat{Y}_0 \mapsto \hat{Y}_{T-\delta}, \iota_\Omega)_\sharp(\gamma \otimes \mathbb{P}) \tag{33}$$

Marginalizing over $\mathbb{P}$, this is a (probability) measure on $\mathbb{R}^d \times \mathbb{R}^d$ since the backward SDE paths are continuous: the backward SDEs admit the following integral formulation, where $h(r, Y_r)$ is the drift term of the backward SDE:

$$Y_t = Y_0 + \int_0^t h(r, Y_r) dr + W_t.$$

Moreover, $\hat{\gamma}$ is a coupling between $Y_{T-\delta} \sim \mu_\delta$ and $\hat{Y}_{T-\delta} \sim \nu_\delta$. We compute:

$$\begin{aligned}
\|\mathbb{E}_{\mu_\delta}[f] - \mathbb{E}_{\nu_\delta}[f]\| = \|\iint \mathbb{E}[f(x) - f(y)] \, d\hat{\gamma}(x, y)\| \\
\leq \iint \mathbb{E}[\|f(Y_{T-\delta}^x) - f(\hat{Y}_{T-\delta}^y)\|] \, d\gamma(x, y) \\
\leq L \iint \|Y_{T-\delta}^x - \hat{Y}_{T-\delta}^y\| \, d\gamma(x, y) \\
\leq CL \iint \|Y_0^x - \hat{Y}_0^y\| \, d\gamma(x, y) \\
= CL \iint \|x - y\| \, d\gamma(x, y) \leq CL \left(\iint \|x - y\|^2 \, d\gamma(x, y)\right)^{1/2}.
\end{aligned}$$

The desired inequality follows by taking infimums over admissible couplings $\gamma \in \Gamma(\mu_T, \nu_T)$. $\quad\square$

## G APPROXIMATE TIMINGS

All times are done on Nvidia A6000 GPUs with 48GB of VRAM. We note that synthesis time as reported in Su et al. (2024); Sun et al. (2024) do not include the time required to generate the latent variables, and thus are not sufficiently representative of the end-to-end time required to distill the dataset.

Table 5: Time required for each step of dataset distillation on ImageNet-1K. Synthesis requires application of the Stable Diffusion V1.5 model to each distilled latent variable, and soft label requires application of the pre-trained ResNet-18 model to each distilled image. Memory usage is constant between IPCs due to equal batch size.

| Step | Time (IPC 10) | Time (IPC 100) |
| --- | --- | --- |
| 1 (Latent clustering) | 8 hours | 8 hours |
| 2 (Synthesis) | 2 hours | 1 day |
| 3 (Soft label) | 1 hour | 16 hours |
| 4 (Training ResNet18) | 2.5 hours | 9 hours |

# H EXPERIMENT HYPERPARAMETERS

We detail the parameters when training the student networks from the distilled data. They are mostly similar to Su et al. (2024).

For consistency and a more direct comparison with previous methods, we use the pre-trained PyTorch ResNet-18 model to compute the soft labels, using the same protocol as Su et al. (2024). After computing the soft labels using the pre-trained ResNet-18 model, we train new ResNet-18, ResNet-50 and ResNet-101 models to match the soft labels. The data augmentation is also identical, so that the only differences are the addition of the weights to the training objective and some minor hyperparameter tuning for the new objective. The latent diffusion model chosen for latent generation and image synthesis is the publicly available pre-trained Stable Diffusion V1.5 model, the same as $D^4M$.

Table 6: Hyperparameter setting for ImageNet-1K experiments.

| Setting | Value |
| --- | --- |
| Network | ResNet |
| Input size | 224 |
| Batch size | 1024 |
| Training epochs | 300 |
| Augmentation | RandomResizedCrop |
| Min scale | 0.08 |
| Max scale | 1 |
| Temperature | 20 |
| Optimizer | AdamW |
| Learning rate | 2e-3 for Resnet18, 1e-3 otherwise |
| Weight decay | 0.01 |
| Learning rate schedule | $\eta_{k+1} = \eta_k/4$ at epoch 250 |

**Variance reduction (heuristic).** We note that the variance of the number of cluster assignments can vary significantly, sometimes up to two orders of magnitude, such as in Figure 2. After normalizing the cluster counts in Step 3 of Algorithm 1 to give weights $w \in (0, 1)$, most weights are very small, and do not contribute much to the neural network training. To reduce this effect, we use the per-centroid weights as follows,

$$w_k^{(L)} = \sqrt{K} \sqrt{v_k^{(L)} / \sum_{j=1}^{K} v_j^{(L)}}. \tag{34}$$

# I ABLATION ON WEIGHTING

To test the effect of the heuristic weights used in (34), we consider the setting of the DiT backbone Section 4.2. We consider three different weightings within the training in Step 4:

1. Constant weights $w \equiv 1$. This is equivalent to $D^4M$.

2. The heuristic weighting strategy (34).

3. The direct cluster weights, given by $w_k \propto v_k^{-1}$, where $v_k$ are the cluster counts from the $k$-means, normalized such that the sum over each class is 1.

We use the same cluster images and only differ the weights. Table 7 details the test accuracy for IPC 10 on ImageNet-1K with various weights, applied with a ResNet-18 teacher and ResNet-18 student architectures. We observe that the heuristic weighting strategy outperforms $D^4M$ across different learning rates. Moreover, the direct cluster weights are able to obtain higher test accuracy, albeit being more sensitive to the choice of learning rate.

Table 7: Ablation across different weighting strategies when training a new student network with soft labels. We observe that the heuristic weight used in the main experiments always outperforms $D_4M$. The direct cluster weighting is more sensitive to learning rate, but eventually outperforms the heuristic weighting for higher learning rates.

| Learning rate | 1e-3 | 2e-3 | 5e-3 | 1e-2 |
|---|---|---|---|---|
| No weighting ($D^4M$) | 52.1 | 52.5 | 51.5 | 52.2 |
| Heuristic weight (34) | 52.3 | 52.6 | 53.2 | 53.6 |
| Direct cluster weighting | 44.1 | 50.3 | 54.6 | 55.6 |

