# OpenReview forum: "Dataset Distillation as Pushforward Optimal Quantization"
_ICLR.cc/2026/Conference — ICLR 2026 Poster_

### Official Review · Reviewer_Pn8q · 2025-10-19

**Soundness:** 4
**Presentation:** 4
**Contribution:** 3
**Rating:** 6
**Confidence:** 4

**Summary:**

The paper reframes disentangled dataset distillation as an optimal quantization problem in latent space, adds per-centroid weights (a small but material tweak), and proves a diffusion-pushforward consistency bound; empirically the paper proposes DDOQ , and it beats D4M and is competitive with DiT-guided methods at higher IPC on ImageNet-1K.

**Strengths:**

1.Casting disentangled DD as optimal quantization ties practice to classical theory; the pushforward bound (Theorem 1) is simple and actionable.

2.Adding weights to latent centroids is near-free and consistently helps (lower W₂; small but real accuracy gains over D4M).

3.Multi-IPC, multiple students (R-18/50/101; MobileNet-V2; EfficientNet-B0; Swin-T) and two generators (LDM, DiT), with cross-teacher/student table.

4.Strong results with DiT. Beats Minimax/IGD on full ImageNet-1K at low IPC; competitive on ImageNette/Woof.

5.Clean pipeline and well-written paper.

**Weaknesses:**

1. The empirical lift mainly comes from adding weights to D4M-style clustering; it’s an elegant repackaging more than a new core mechanism. A D4M+weights control would isolate weighting vs. barycenter vs. quantizer effects.

2. Theorem 1 requires δ>0 (no bound at exact image time), compact support, and Lipschitz test functionals; constants depend on (δ,T,R,d). Useful, but the gap to practical (non-Lipschitz) training signals remains.

3. Teacher dependence and fairness. Main UNet/LDM tables fix ResNet-18 soft labels (cap at 69.8), but teacher swaps materially affect generalization (Table 3) and DiT tables use different baselines. A teacher-swap on ImageNet-1K main table would de-bias conclusions.

4. Results focus on accuracy; there’s little about wall-clock/GPU hours for distillation + training across methods/backbones. (RDED omitted at high IPC due to compute, another signal that cost matters.)

5. DDOQ underperforms D4M for Swin-T students at IPC50; this suggests hyperparameter or representation-mismatch issues that deserve diagnosis.

**Questions:**

1. What exactly drives the gains, weights or quantizer? Please add a D4M + weights variant and a DDOQ w/o weights ablation to separate effects.

2. What is the effective latent dimension d used by LDM/DiT in your pipeline, and how do W₂ and accuracy scale with K (log–log plots) to test the K⁻¹/ᵈ prediction?

3. Please replicate Table 2 with a ViT-B and Swin-T soft-label teacher to test robustness of conclusions beyond ResNet-18’s 69.8 cap.

4. Report end-to-end cost (distill + train) for D4M vs DDOQ vs guidance-based baselines, at IPC10/50/200, on fixed hardware.

5. Any insights (e.g., label entropy, weight dispersion, LR/BS sensitivity) on why Swin-T lags with DDOQ in Table 3?

---

> ### Author Response · Authors · 2025-11-17
>
> We thank the reviewer for the positive feedback regarding this work. Please find responses to each of the reviewer's concerns and questions below.
>
> > The empirical lift mainly comes from adding weights to D4M-style clustering; it’s an elegant repackaging more than a new core mechanism. A D4M+weights control would isolate weighting vs. barycenter vs. quantizer effects.
>
>  As noticed by the reviewer, the algorithmic difference is small, but was discovered by going through the theoretical lifting. As far as we are aware, the clustering-based approach is the only framework that allows for consistent dataset distillation in the finite IPC setting; the bi-level approaches would likely require some assumptions on the risk (that have not been explored in the literature), whereas the diffusion guidance models appear to sample from a modified data distribution as a result of the heuristics.
>
>  We note that the cluster centroids in D4M and DDOQ are identical since they both arise from k-means clustering, and that the weights are the mechanism through which we get a lower $W_2$ distance. We will add some text to mention this.
>
> > Theorem 1 requires δ>0 (no bound at exact image time), compact support, and Lipschitz test functionals; constants depend on (δ,T,R,d). Useful, but the gap to practical (non-Lipschitz) training signals remains.
>
> We note that $\delta>0$ is a typical assumption as in Remark 2, due to blowup at time $\delta=0$, a consequence of the manifold hypothesis; the diffusion is not defined at time 0. Compact support is also a typical assumption, see e.g. [Chen19] and the related literature in approximation theory. The listed constants are minimal such that a meaningful bound holds, and we give the constant explicitly in Eq. (32).
>
>  While we use the smaller $W_2$ distance to informally argue that gradient steps (w.r.t. network parameters) are "closer to the true gradient step" in the paragraph after Theorem 1, we agree that the application to real neural networks requires additional work. A possible method is to consider the gradient step in an appropriate function space, but we are not aware of any literature that does this for finite networks.
>
> > Teacher dependence and fairness. Main UNet/LDM tables fix ResNet-18 soft labels (cap at 69.8), but teacher swaps materially affect generalization (Table 3) and DiT tables use different baselines. A teacher-swap on ImageNet-1K main table would de-bias conclusions.
>
> We note that using ResNet-18 as a teacher for ImageNet-1K appears to be standard in the literature. Nonetheless, the figures listed are fair towards each method, as they use the same teacher and student models. We may try to add this for IPC10 if time permits, but this would entail a separate set of hyperparameter tuning for each of the compared baselines as well (since such teacher transfers are not reported for ImageNet-1K).
>
> > Results focus on accuracy; there’s little about wall-clock/GPU hours for distillation + training across methods/backbones. (RDED omitted at high IPC due to compute, another signal that cost matters.)
>
>  We are unable to find explicit times for the diffusion guidance methods Minimax and IGD. Minimax simply states "it takes less than 1 hour to distill 100IPC for a 10-class ImageNet subset", and IGD states "it takes about 5-6$\times$ the time of the vanilla (generative model) method". In contrast, our method is split into clustering and forward pass. Our clustering + synthesis takes about 30 hours for 100IPC for 1000-class ImageNet (in Appendix G), suggesting it is still faster than the minimax and IGD methods, which would take around 100 hours if extrapolating from their statements.
>
>  Compared to D4M, DDOQ incurs trivial computational overhead, we refer to Table 6 in the D4M paper for per-iteration cost. RDED does not have any generative component so is faster; however, the generated images do not lie around the data manifold, due to the patch stitching.
>
> > DDOQ underperforms D4M for Swin-T students at IPC50; this suggests hyperparameter or representation-mismatch issues that deserve diagnosis.
>
>  We did a similar hyperparameter search to the other architectures (3 choices of learning rate and scheduler) based on the parameters in D4M, but these generally appear to be more difficult to come by (For example, IGD does not provide hyperparameter choices for training a Swin-Transformer student architecture.)
>
>  After some brief literature review, we found that training Swin-T is also quite sensitive to other optimizer hyperparameters, such as augmentation strategy, weight decay, gradient clipping etc.

---

> > ### Author Response · Authors · 2025-11-17
> >
> > > What exactly drives the gains, weights or quantizer? Please add a D4M + weights variant and a DDOQ w/o weights ablation to separate effects.
> >
> >  The gains are given by both weights and quantizing (as opposed to random sampling). As demonstrated in Table 1, the Wasserstein distance between the latent points is reduced after adding the weights. Using quantization also drops the Wasserstein distance compared to random sampling (DiT column in Table 4), yielding better training.
> >
> > > What is the effective latent dimension d used by LDM/DiT in your pipeline, and how do W₂ and accuracy scale with K (log–log plots) to test the $K^{-1/d}$ prediction?
> >
> >  The latent dimension of the encoder is $32 \times 32 \times 4$. However, using a two-point regression on the Wasserstein distances in Table 1 suggests that the intrinsic dimension is roughly $d\approx 64$. One reference estimates the intrinsic dimension of ImageNet to be roughly $d \approx 43$ [Pope21], so we are not far off from this estimation. We note that since we are in the finite particle setting, we will get zero error once the number of centroids is equal to the number of training points.
> >
> > > Please replicate Table 2 with a ViT-B and Swin-T soft-label teacher to test robustness of conclusions beyond ResNet-18’s 69.8 cap.
> >
> >  We will do this if there is time after the other cheaper experiments as in the general comment. We note that the currently reported baselines are given in their respective papers, so such an experiment would necessarily entail running their methods with the modified teachers as well.
> >
> > > Report end-to-end cost (distill + train) for D4M vs DDOQ vs guidance-based baselines, at IPC10/50/200, on fixed hardware.
> >
> > Our (and equivalently D4M's) costs are given in Appendix G; synthesis and training times are linear in IPC. Clustering time is bottlenecked by application of the encoder, so is effectively constant in this scenario. We were unable to find concrete times for the guidance-based baselines in their papers, nor a discussion on the scaling of their costs with respect to IPC.
> >
> > > Any insights (e.g., label entropy, weight dispersion, LR/BS sensitivity) on why Swin-T lags with DDOQ in Table 3?
> >
> > See response to point 5. Perhaps the relationship between "weights being close" and "function being close" is different for the transformer architecture, compared to CNN architectures (which may possibly benefit from random matrix results to reduce the effective Lipschitz constant of the function w.r.t. parameters).
> >
> > [Pidstrigach22] Jakiw Pidstrigach. Score-based generative models detect manifolds. Advances in Neural Information Processing Systems, 35:35852–35865, 2022
> >
> > [Chen19] Chen, M., Jiang, H., Liao, W., & Zhao, T. (2019). Efficient approximation of deep relu networks for functions on low dimensional manifolds. _Advances in neural information processing systems_, _32_.
> >
> > [Pope21] Pope, P., Zhu, C., Abdelkader, A., Goldblum, M., & Goldstein, T. (2021). The intrinsic dimension of images and its impact on learning. ICLR '21.

---

### Official Review · Reviewer_98qT · 2025-10-26

**Soundness:** 2
**Presentation:** 1
**Contribution:** 2
**Rating:** 2
**Confidence:** 4

**Summary:**

This paper proposes Dataset Distillation by Optimal Quantization (DDOQ), which reframes disentangled dataset distillation as an optimal quantization problem. The method leverages latent diffusion models (LDMs) to represent data distributions in a low-dimensional latent space, performs weighted clustering (CLVQ) to approximate the latent data measure, and uses the generative decoder to reconstruct synthetic training samples.

**Strengths:**

1. The paper introduces a new conceptual connection between dataset distillation and optimal quantization under the Wasserstein distance framework.
2. The authors propose to use the Wasserstein distance between the distilled latent representations and the original latent data distribution as a quantitative indicator of how well the synthetic dataset approximates the real data distribution.
3. The proposed method, DDOQ, improves performance in higher images-per-class (IPC) settings.

**Weaknesses:**

1. The introduction looks like a related work and fails to clearly introduce the research gap and specific contributions of the paper. As a result, it is difficult for readers to understand what is novel about this work.
2. The proposed DDOQ method is a modification of D4M, replacing uniform clustering with a weighted k-means step. The overall pipeline (latent clustering, diffusion decoding, and weighted training) remains conceptually similar to previous methods. The framing of the approach as “optimal quantization” seems more like a theoretical reinterpretation than a fundamentally new algorithmic contribution.
3. In Section 3.1, it is not explicitly shown how optimal quantization principles are concretely applied to dataset distillation. The practical novelty seems the introduction of weights, but these weights are simply normalized cluster weights (as shown in Algorithm 1 in line 278) and not directly derived from optimal quantization theory.
4. The paper is mathematically dense and lacks intuitive explanation or visual figures. A framework diagram or flowchart illustrating the DDOQ pipeline would greatly help readers grasp how the theoretical formulation connects to the practical implementation.
5. In Table 2, the caption claims that “the maximum performance for all methods should be 69.8,” but this number does not appear in the table, leading to confusion. Additionally, the DDOQ performance at IPC=10 is substantially worse than RDED, indicating that the proposed method may not generalize well across different IPC settings.
6. The paper lacks a detailed ablation study analyzing the effects of key hyperparameters, such as the latent dimension, the number of quantization points K, and the weighting scheme.

**Questions:**

Could the authors provide a table comparing the computational cost (e.g., memory usage, and GPU hours) of DDOQ against other baseline methods such as D4M and RDED?

---

> ### Author Response · Authors · 2025-11-17
>
> We thank the reviewer for their comprehensive suggestions regarding rewriting the paper in a clearer fashion, which we may address with the additional page. Please find responses to the reviewer's concerns below.
>
> We would like to clarify that the second strength listed by the reviewer is not a core point of this work. The Wasserstein distance is introduced as a distance between probability distributions, correlating to how well a gradient step under the distilled data will correspond to a gradient step under the true training data. Table 1 (on Wasserstein distances) is simply a sanity check that approximating the weights indeed decreases this distance, as opposed to keeping the weights uniform as in D4M.
>
> > The introduction looks like a related work and fails to clearly introduce the research gap and specific contributions of the paper. As a result, it is difficult for readers to understand what is novel about this work.
>
>  The novelty is presented in the first bullet point in the introduction. In particular, we provide a theoretical justification for using clustering for dataset distillation. We note that the competing methods, namely bi-level methods and diffusion guidance methods, have not been shown to converge to the underlying data distribution as the number of distilled points increases. We conjecture that for the bi-level methods, additional assumptions on the risk $\mathcal{R}$ are required; while for the diffusion guidance methods, the underlying modified objective means that the distilled points will converge to a different data distribution. We will rewrite the introduction to clarify the significance and position of our work in the wider literature.
>
> > The proposed DDOQ method is a modification of D4M, replacing uniform clustering with a weighted k-means step. The overall pipeline (latent clustering, diffusion decoding, and weighted training) remains conceptually similar to previous methods. The framing of the approach as “optimal quantization” seems more like a theoretical reinterpretation than a fundamentally new algorithmic contribution.
>
>  Similarly to the above response, the pipeline is similar because this is the only consistent dataset distillation method that we found (where consistency is defined as converging to the underlying data distribution as the number of distilled samples increases). The modification is directly motivated from this observation.
>
> > In Section 3.1, it is not explicitly shown how optimal quantization principles are concretely applied to dataset distillation. The practical novelty seems the introduction of weights, but these weights are simply normalized cluster weights (as shown in Algorithm 1 in line 278) and not directly derived from optimal quantization theory.
>
>  We refer to Proposition 3 to interpret the weights: it states that the normalized cluster weights converge to the measure of the optimal Voronoi cells. The measures of the Voronoi cells are shown to be the optimal weights in Proposition 1.
>
> > The paper is mathematically dense and lacks intuitive explanation or visual figures. A framework diagram or flowchart illustrating the DDOQ pipeline would greatly help readers grasp how the theoretical formulation connects to the practical implementation.
>
>  We will add a diagram and additional explanation on the overall flow of the work. We hope that the above exposition aids the reviewer to understand the presented theoretical perspective on dataset distillation.
>
> > In Table 2, the caption claims that “the maximum performance for all methods should be 69.8,” but this number does not appear in the table, leading to confusion. Additionally, the DDOQ performance at IPC=10 is substantially worse than RDED, indicating that the proposed method may not generalize well across different IPC settings.
>
> We will add a column to the right side of Table 2 to more clearly indicate the maximum performance.
>
> RDED performs surprisingly well at IPC=10 in this case because the generated images are not similar to the natural image distribution, instead, they are made of a 2x2 grid of images (see lines 416-420). This appears to suit convolutional architectures but may fail for non-convolutional architectures like transformers (RDED does not test with transformer-based student models). It is also unclear what distribution RDED is actually sampling from as the number of images per class increases, since the patch-based objective is purely heuristic.

---

> > ### Author Response · Authors · 2025-11-17
> >
> > > The paper lacks a detailed ablation study analyzing the effects of key hyperparameters, such as the latent dimension, the number of quantization points K, and the weighting scheme.
> >
> >  The latent dimension of the pretrained diffusion models is $32 \times 32 \times 4$. In practice, the $d$ should be the intrinsic dimension of the data. Using a two-point regression on the Wasserstein distances in Table 1 yields that the dimension is roughly $d \approx 64$, which is not too far from the estimate of 43 [Pope21] in the literature.
> >
> > The number of quantization points $K$ is simply the number of images per class (IPC). We quantize per-class, as typical in distillation literature.
> >
> >  We will add an ablation on the weighting scheme.
> >
> > > Could the authors provide a table comparing the computational cost (e.g., memory usage, and GPU hours) of DDOQ against other baseline methods such as D4M and RDED?
> >
> >  Approximate times are given in Appendix G. Otherwise, since the computational overhead of our method over D4M is trivial, the computational time is the same as D4M. As stated in their paper, the SOTA baseline IGD takes around 6 times the time of a usual generative pass to generate an image, while RDED does not involve any generative process to reduce their generation time. The memory usage during clustering and distillation is simply that of the encoder and decoder portions respectively of the latent diffusion model.
> >
> > [Pope21] Pope, P., Zhu, C., Abdelkader, A., Goldblum, M., & Goldstein, T. (2021). The intrinsic dimension of images and its impact on learning. ICLR '21.

---

### Official Review · Reviewer_Zzhq · 2025-10-30

**Soundness:** 4
**Presentation:** 3
**Contribution:** 4
**Rating:** 6
**Confidence:** 3

**Summary:**

The paper aims to improve representative selection for classes in dataset distillation. It reframes the selection process as an optimal quantization problem and supports the approach with theoretical proofs and experimental validation. The objective is to provide a more principled and theoretically grounded way of selecting better representatives for dataset distillation for sample efficient training. This combination of theory and empirical study gives the work potential value for the dataset distillation research.

**Strengths:**

1. Frames the dataset distillation problem as an optimal quantization problem and supports the suggested improvements with appropriate theoretical proofs. The theoretical formalization connects optimal quantization error with downstream expectation differences in the image domain, which helps bridge diffusion-based priors and representative selection in a mathematically grounded way.

2. Theorem 1 gives an explicit upper bound for the Wasserstein distance in latent space, which they show is related to the expectation difference in image-space functions after reverse diffusion. This link between latent space distance and image-space fidelity is useful and not previously formalized for these disentangled distillation methods.

3. The complexity of the proposed method is low, but the performance gains are consistent and meaningful compared to baselines.

4. Experiments covering other approaches for the same latent diffusion backbone and comparison of different prior selection methods for DiT are good to show that the gains are from the proposed approach and translate to other backbones.

**Weaknesses:**

1. The theoretical foundation simply builds upon existing proofs from quantization theory, so the novelty in theoretical contributions is limited. The actual improvements suggested are limited and incremental, as operationally, it simply adds weights to cluster centers.

2. A weighted loss is used while training the student models, which makes it unclear if the improvements are from better diffusion prior selection via clustering or better optimization of student model via weighted loss. An ablation study clarifying this will improve the contribution.

**Questions:**

1. The bad performance of the Swin-T transformer as a student is attributed to hyperparameters. Providing results with optimal hyperparameters will be helpful to identify if transformer architecture poses some limitation to this work or disentangled distillation in general.

2. Proposition 1 and 2 use the notation δ_{x_i}. Please clarify whether this denotes a Dirac delta function (i.e., a unit point mass at x_i) or some other function indicating the Voronoi cell’s probability mass.

---

> ### Author Response · Authors · 2025-11-17
>
> We thank the reviewer for the positive review and appreciation of the theoretical component. We address the reviewer's questions below.
> > The theoretical foundation simply builds upon existing proofs from quantization theory, so the novelty in theoretical contributions is limited. The actual improvements suggested are limited and incremental, as operationally, it simply adds weights to cluster centers.
>
>  We agree that the end algorithmic change is small, but still lead to quantifiable performance gains. We believe that it is important to motivate the importance of introducing the cluster weights, which was discovered by going through the optimal quantization interpretation.
>
> > A weighted loss is used while training the student models, which makes it unclear if the improvements are from better diffusion prior selection via clustering or better optimization of student model via weighted loss. An ablation study clarifying this will improve the contribution.
>
> See common response.
>
> > The bad performance of the Swin-T transformer as a student is attributed to hyperparameters. Providing results with optimal hyperparameters will be helpful to identify if transformer architecture poses some limitation to this work or disentangled distillation in general.
>
> We did a similar hyperparameter search to the other architectures (3 choices of learning rate and scheduler) based on the parameters in D4M, but these generally appear to be more difficult to come by (For example, IGD does not provide hyperparameter choices for training a Swin-Transformer student architecture.)
>
>  After some brief literature review, we found that training Swin-T is also quite sensitive to other optimizer hyperparameters, such as augmentation strategy, weight decay, gradient clipping etc.
>
> > Proposition 1 and 2 use the notation $\delta_{x_i}$. Please clarify whether this denotes a Dirac delta function (i.e., a unit point mass at x_i) or some other function indicating the Voronoi cell’s probability mass.
>
> $\delta(x_i)$ refers to the Dirac mass at $x_i$, while $\mu(C_i)$ denotes the mass of the Voronoi cells. We will clarify this in Proposition 1.

---

### Official Review · Reviewer_gQzu · 2025-10-31

**Soundness:** 3
**Presentation:** 3
**Contribution:** 3
**Rating:** 6
**Confidence:** 5

**Summary:**

This paper provides a theoretical interpretation of dataset distillation (DD) by framing it as an optimal quantization problem in latent space. The authors show that modern disentangled DD methods can be understood as approximating the true data distribution via finite quantization of its latent representation.
Building on these observations, they propose Dataset Distillation by Optimal Quantization (DDOQ), which performs weighted clustering (via CLVQ) in the latent space of a pretrained diffusion model and uses the resulting centroids and weights to reconstruct distilled images. Empirically, DDOQ achieves improved performance over prior works such as D4M and RDED on ImageNet-1K, ImageNette, and ImageWoof, with notable advantages in medium IPC regimes and better inter-model generalization.

**Strengths:**

- **Theoretical contribution**: The key theoretical contribution is the formal link between quantization theory, Wasserstein distance, and dataset distillation consistency. Theorem 1 demonstrates that score-based diffusion preserves the distributional closeness between the raw data and its quantized latent approximation, resulting in consistent gradient expectations during training on the distilled dataset. Corollary 1 further establishes asymptotic convergence rates $\mathcal{O}(K^{-1/d})$ for the approximated data distribution in image space, offering the consistency guarantees for diffusion-based DD methods.
- **Clean problem framing**: Reformulating dataset distillation as a pushforward optimal quantization problem is an insightful contribution that bridges classical quantization theory and modern generative modeling.
- **Comprehensive experiments**: Evaluation covers both UNet and DiT-based diffusion models, multiple IPC regimes, and cross-architecture generalization. The results are consistent and competitive with SOTA baselines.

**Weaknesses:**

- **Heuristic implementation > theoretical analysis**: The core theoretical argument is that optimal quantization (non-uniform weights) is superior to finding the Wasserstein barycenter (uniform weights). However, the final implementation does not utilize these theoretically derived weights. Instead, it employs an ad-hoc heuristic (Eq. 34) output for "variance reduction." This choice is not justified by the theory and is not ablated. This undermines the central claim that the theory guides the method's design. Rather, the theory appears to be more of a post-hoc justification for the general idea of weighting, while the actual implementation relies on an unexplained heuristic.
- **Lack of practical guidance**:
    - The paper introduces weighting as a core contribution but provides little analysis of its behavior, interpretability, or sensitivity during training.
    - The bound in Theorem 1 contains a constant C with dependencies that are not analyzed in a practical context (e.g., how to choose diffusion time T to optimize the bound). The convergence rate $\mathcal{O}(K^{-1/d})$ is standard for quantization but does not lead to new insights on how to choose an optimal latent dimension d or number of prototypes K for a given budget.

**Questions:**

- **More ablations**: Could the authors provide an ablation study on ImageNet-1K (e.g., at IPC 10) that directly compares: (a) no weights (D4M baseline), (b) the theoretically motivated weights, and (c) your proposed heuristic weights? Without these ablations, it is difficult to assess whether the theory is useful in practice or if an arbitrary heuristic is the true source of improvement.
- **Source of performance gains**: Can you quantify the relative error reduction from (a) moving from D4M to DDOQ on a fixed U-Net backbone, versus (b) moving from a U-Net to a DiT backbone using a fixed D4M algorithm? This would help disentangle the contribution of your method from the contribution of the stronger (or different) generative model.
- **OQ utility**: Beyond justifying the use of weights, does the OQ perspective offer any other suggestions for concrete algorithmic improvements? For instance, the $\mathcal{O}(K^{-1/d})$ error rate suggests a trade-off between the number of prototypes K and latent dimension d. Does your theory provide any guidance on how to choose d for a given distillation budget K? If not, what is the practical utility of the theory beyond re-describing weighted k-means?
- **Necessity of weighting in high-dimensional latent spaces**: The paper suggests that the weighting scheme mitigates the mismatch between the limited number of quantization points (IPC) and the high dimensionality of the latent space. Could the authors elaborate on this relationship quantitatively? For instance, is there an empirical or theoretical threshold linking the number of latent samples K and latent dimension d beyond which weighting becomes less critical (e.g., K>>d)?
- **Higher IPC regimes**: If the number of distilled images increases substantially (e.g., IPC@1000), do the authors expect the weighting adjustment to remain influential, or would the raw cluster frequencies already approximate the latent density sufficiently? Clarifying this could help understand whether the proposed weighting primarily benefits low-data regimes or remains beneficial asymptotically.

---

> ### Author Response · Authors · 2025-11-17
>
> We thank the reviewer for the positive review and detailed feedback regarding our work. We address the reviewer's questions below.
> > **Heuristic implementation > theoretical analysis**: The core theoretical argument is that optimal quantization (non-uniform weights) is superior to finding the Wasserstein barycenter (uniform weights). However, the final implementation does not utilize these theoretically derived weights. Instead, it employs an ad-hoc heuristic (Eq. 34) output for "variance reduction." This choice is not justified by the theory and is not ablated. This undermines the central claim that the theory guides the method's design. Rather, the theory appears to be more of a post-hoc justification for the general idea of weighting, while the actual implementation relies on an unexplained heuristic.
>
> We will add an ablation on this. The heuristic can be roughly interpreted as somewhere in between the uniform weights and the optimal quantization weights (an $\ell_2$ interpolation between the $\ell_1$-normalized weights and the uniform weights).
> >  **Lack of practical guidance**:
> >    - The paper introduces weighting as a core contribution but provides little analysis of its behavior, interpretability, or sensitivity during training
>
> We will try to add a small ablation regarding the sensitivity to the choice of weights.
> >   - The bound in Theorem 1 contains a constant C with dependencies that are not analyzed in a practical context (e.g., how to choose diffusion time T to optimize the bound). The convergence rate $\mathcal{O}(K^{-1/d})$ is standard for quantization but does not lead to new insights on how to choose an optimal latent dimension d or number of prototypes K for a given budget.
>
>  - The choice of diffusion time for diffusion models is an open question as far as we are aware. The literature typically picks a $T$ that is "good enough" to approximate the terminal Gaussian noise distribution of the forward SDEs.
>
>  - Choosing these optimal hyperparameters could be an interesting open problem but is outside the scope of this work, which already tries to find the optimal distilled training points. Finding the minimum $K$ to achieve say 90\% of the maximum accuracy could likely be achieved using active learning techniques or another fine-tuning technique.
> > **More ablations**: Could the authors provide an ablation study on ImageNet-1K (e.g., at IPC 10) that directly compares: (a) no weights (D4M baseline), (b) the theoretically motivated weights, and (c) your proposed heuristic weights? Without these ablations, it is difficult to assess whether the theory is useful in practice or if an arbitrary heuristic is the true source of improvement.
>
>  See common response.
>
> > **Source of performance gains**: Can you quantify the relative error reduction from (a) moving from D4M to DDOQ on a fixed U-Net backbone, versus (b) moving from a U-Net to a DiT backbone using a fixed D4M algorithm? This would help disentangle the contribution of your method from the contribution of the stronger (or different) generative model.
>
> See common response. We note that we may not be able to fully ablate additional hyperparameters, e.g. guidance strength for D4M using the DiT weights, as opposed to the presumably tuned best performances provided in the published works.

---

> > ### Author Response · Authors · 2025-11-17
> >
> > > **OQ utility**: Beyond justifying the use of weights, does the OQ perspective offer any other suggestions for concrete algorithmic improvements? For instance, the $\mathcal{O}(1/K)$ error rate suggests a trade-off between the number of prototypes K and latent dimension d. Does your theory provide any guidance on how to choose d for a given distillation budget K? If not, what is the practical utility of the theory beyond re-describing weighted k-means?
> >
> >  We would like to clarify that the latent dimension $d$ corresponds to the intrinsic dimension of the data and is not a variable. However in practice, since $d$ is intractable for general datasets, it can be upper bounded by the latent dimension of the space. Performing a simple two-point regression with Table 1 suggests that $d\approx 64$, where increasing the number of points by a factor of 5 means a decrease of about $2.5\%$ in the $W_2$ distance. This is a bit higher than another intrinsic dimension estimation of 43 [Pope21].
> >
> >  The theory most importantly states that a clustering-based distillation approach is consistent as the number of samples increases. As far as we are aware, we are the first work to reformulate dataset distillation in such a consistent manner. In fact, many of the heuristic works mentioned can not be shown to be consistent in the limit -- for example, the bi-level methods mentioned in the introduction would require some additional assumptions on the risks, while patch-based methods like RDED and the recent diffusion guidance models fundamentally alter the target data distribution. D4M is the only work we found that is able to be interpreted in the finite sample case with corresponding asymptotic results -- we explicitly interpret this and find the missing weights for the samples.
> >
> > > **Necessity of weighting in high-dimensional latent spaces**: The paper suggests that the weighting scheme mitigates the mismatch between the limited number of quantization points (IPC) and the high dimensionality of the latent space. Could the authors elaborate on this relationship quantitatively? For instance, is there an empirical or theoretical threshold linking the number of latent samples K and latent dimension d beyond which weighting becomes less critical (e.g., K>>d)?
> >
> >  This question appears to ask what the constants are in the $\mathcal{O}(K^{-1/d})$ convergence, with and without the weights. As demonstrated, weighting is always better than not weighting. While the weighted case has explicit constants depending on the moments [Liu20], we are not aware of any works that characterize the asymptotic constants if using un-weighted clusters, even in the Gaussian case. The constant weights case however can be upper-bounded, using the distance of the empirical measure with randomly sampled points. A possible gap would depend on the variance of the volumes of the Voronoi cells, however the interaction of these volumes with the points is complicated.
> >
> > > **Higher IPC regimes**: If the number of distilled images increases substantially (e.g., IPC@1000), do the authors expect the weighting adjustment to remain influential, or would the raw cluster frequencies already approximate the latent density sufficiently? Clarifying this could help understand whether the proposed weighting primarily benefits low-data regimes or remains beneficial asymptotically.
> >
> >  In the higher IPC regimes, say IPC1000, the size of the distilled dataset would become quite close to the size of the original dataset. For ImageNet, each class has around 1200 images. Therefore, the error would be dominated by the deviation of the sample distribution to the true data distribution. An exact characterization of this is outside the scope of our work, however we can point the reviewer to some recent references on inexact score matching [Pedrotti24].
> >
> >  To answer the reviewer's questions on whether the weights are necessary for convergence in the asymptotic regime, they are not. Classical MCMC literature [Fournier15, Thm. 15] shows that the Wasserstein-2 distance also decays as $\mathcal{O}(K^{-1/d})$, with similar results when replacing $2$ with different $p$ depending on the Sobolev embeddings. However, the inclusion of the weights makes the Wasserstein-2 estimations strictly better in the non-asymptotic regime. We note that [Liu20] provides different bounds on the expected Wasserstein-2 distance of the clusters.
> >
> > [Pope21] Pope, P., Zhu, C., Abdelkader, A., Goldblum, M., & Goldstein, T. (2021). The intrinsic dimension of images and its impact on learning.
> >
> > [Fournier15] Fournier, N., & Guillin, A. (2015). On the rate of convergence in Wasserstein distance of the empirical measure.
> >
> > [Pedrotti24] Pedrotti, F., Maas, J., & Mondelli, M. Improved Convergence of Score-Based Diffusion Models via Prediction-Correction.
> >
> > [Liu20] Liu, Y., & Pages, G. (2020). Convergence rate of optimal quantization and application to the clustering performance of the empirical measure.

---

### Author Response · Authors · 2025-11-17
**Response to common concerns**

**General comments**

We thank the reviewers for their time, detailed reviews, and constructive feedback. There are some common questions between the reviewers that we address here.

* **Importance on the heuristic weighting scheme**. We will add a short ablation here regarding the use of Equation 34, versus a direct application of the learned weights. In general, since some of the weights are very small for low IPC (quantifying the tails of the distribution), the training becomes more sensitive to step-size. Moreover, the norm of the weights may differ in between batches. For example, if the minibatch contains many highly weighted datapoints, the batch gradient will be larger than compared to if the minibatch contains many low weight datapoints. This can lead to some instability if the weights are too different, which we did not observe with our weighting scheme.
* **Running D4M with the DiT backbone.** We note that D4M using the transformer architecture has not been done in the literature, and existing literature (including the SOTA diffusion-guidance methods) compares with D4M applied using the older U-Net backbone. We will try to run the D4M experiments again with DiT to better ablate the effect of the weights.

### 29 Nov edit:

We thank the reviewers for their patience regarding the new experiments. In view of the recent announcement regarding the OpenReview bug and restriction on reviewer comments, we hope that the below experiments may still address the remaining concerns the reviewers have, and aid the AC's decision.

**New experiment: Ablation on weighting scheme.**
As indicated in our previous response, the heuristic weighting scheme is used to stabilize training. The table below shows the performance of the various methods, using the stronger DiT backbone to generate the images and weights. D4M uses the weight $w \equiv 1$, the heuristic weighting is given by Equation (34), and the direct cluster weighting is given by $w_k \propto v_k^{-1}$, where $v_k$ is the weight of the cluster in $k$-means, and the proportionality constant is such that the class-wise weight sums to 1. We did not tune other hyperparameters.

Observe that the direct cluster weighting is able to outperform D4M as well as the heuristic weighting for larger stepsizes. The heuristic weighting still outperforms D4M (as suggested by Table 2 for U-Net backbone). We will add this table to the appendix.

| Stepsize                 | 1e-3  | 2e-3  | 5e-3  | 1e-2 |
|--------------------------|-------|-------|-------|------|
| No weighting (D4M)       | 52.13 | 52.51 | 51.46 | 52.17 |
| Heuristic weighting      | 52.29 | 52.57 | 53.23 | 53.58 |
| Direct cluster weighting | 44.07 | 50.28 | 54.60 | 55.61 |

**Formatting changes**
As suggested by Reviewer 98qT, we have made the following changes to the manuscript to make it more suitable for a wider audience. Changes in the manuscript are highlighted in blue.
* An additional figure (Figure 1) has been added to page 2 to illustrate the method and the contributions of the work. A reference to Figure 1 is provided on page 7, after the sketch of the proposed DDOQ method.
* The contributions section on page 3 has been slightly rewritten to emphasize the theoretical contribution of this work to the dataset distillation literature.
* Proposition 1 now clarifies the $\delta(x_i)$ notation.
* The start of Section 3 now explicitly states that the weights in the proposed algorithm are automatically determined.
* A comment has been added to the conclusion to mention that similar convergence results for bi-level or diffusion guidance methods are still open problems.

---

> ### Author Response · Authors · 2025-11-30
>
> **Additional comments on the transformer teacher**
> For the interested reviewer, Reviewer Pn8q suggested to replicate Table 2 (LDM with U-Net backbone across 4 different IPCs) with a Swin-T or ViT-B teacher, as opposed to ResNet-18. This is too expensive, so we instead considered to try replicating Table 3 using the DiT backbone and the Swin-T/ViT-B-16 teacher. We report our attempts to remedy the big drop in performance when swapping the ResNet-18 soft-label teacher with a Swin-T teacher. As far as we are aware, using a transformer teacher for soft-labeling on ImageNet-1K has not been explored outside of D4M and our work.
>
> So far, we have tried various hyperparameters in the IPC 10 level, with a ResNet-18 student model, but the maximum accuracy we could get with and without weighting is around 13\% test accuracy, around half of the performance reported in Table 3 (which is IPC 50 instead of 10). We have observed the following issues within the training formulation:
>
> * Since the soft-label training computes the KL distance between the log-softmax probabilities of the teacher and student models, the scale of the network outputs matter. A constant scaling is applied to the outputs of both models before applying the KL loss, referred to as the temperature. The network outputs (before a softmax layer) are the outputs of a linear layer, and we observed that there is a large difference between Swin-T and ResNet-18 outputs.
>
> * In particular, we observe that the $\ell_2$ norm of the output of the (pretrained) Swin-T model is roughly 25, while the $\ell_2$ norm of the (pretrained) ResNet-18 model is roughly 80. (The average norm of an untrained Swin-T's output is around $13$ , and the average norm of an untrained ResNet-18's output is around $22$.).
>
> * The variance of the outputs (as a vector in $\mathbb{R}^{1000}$) is around 0.54 for the pre-trained Swin-T, and around 6.92 for the pre-trained ResNet-18. The variances for an untrained Swin-T are around 0.18 and around 0.45 for an untrained ResNet. The values are summarized in the following tables:
>
> | Output norm                 | ResNet-18  | Swin-T  | ViT-B  |
> |--------------------------|-------|-------|-------|
> | Untrained      | 22 | 13 | 0 |
> | Trained     |  80| 25 | 23 |
>
> | Output variance                 | ResNet-18  | Swin-T  | ViT-B  |
> |--------------------------|-------|-------|-------|
> | Untrained      | 0.45 | 0.18 | 0 |
> | Trained     | 6.92 | 0.54  | 0.53 |
>
> Observe that there is a very big gap between the norms and variances of the pretrained ResNet-18 and pretrained transformer architectures. The output norms and variances are around 20 and 0.4 respectively for the pretrained ResNet-50 and pretrained ResNet-101 networks as well, and explode for the untrained ResNet-50 and ResNet-101 networks.
>
> After finishing the above ablation table, we are now running new experiments to see if we can address this gap. We hope to update this thread with some additional findings before the end of the author rebuttal period.
>
> Dec 3 update: we attempted to rectify this by changing the batch-size, learning rate, and by scaling the soft-labeling loss to make the variance of the student equal to that of the teacher. We were able to obtain a maximum of around 15.4\% test accuracy for DDOQ and around 14.7\% with D4M, using Swin-T as a teacher and ResNet-18 as a student on IPC10.
>
> **Additional comments on the transformer student**
>
> The above table may also be one of the reasons why Swin-T does not perform as well when used as a student architecture. We list some other papers that use transformers as a student architecture on ImageNet-1K: Curriculum Dataset Distillation (57.9 -> 39.2 @ IPC50, Swin-T), CDA (53.4 -> 31.95 @ IPC50, DeiT-Tiny).
>
> A concurrent work "Rectified Decoupled Dataset Distillation" reports an increase in performance of D4M for the Swin-V2-T architecture (60.2 -> 62.2 @ IPC50), but their code is not available. Morerover, this work reports that the transformer-based architectures Swin-V2-T and ViT-B-16 are particularly poor at IPC 10, attaining at most around 22\% accuracy.
>
> We conjecture that techniques from knowledge distillation may be able to assist this gap, such as [Joshi25]
>
> [Joshi25] Joshi, Siddharth, Jiayi Ni, and Baharan Mirzasoleiman. "Dataset Distillation via Knowledge Distillation: Towards Efficient Self-Supervised Pre-Training of Deep Networks." ICLR '25.

---

### Meta-Review · Area_Chair_JvhA · 2026-01-10

**Summary:**

Upon my initial reading of this paper, I found the premise quite intriguing, particularly the authors' claim regarding the resolution of cross-architecture generation issues. However, after a thorough examination of the manuscript and the experimental results, I must express some reservations regarding the execution. First, reliance on ImageNet-1K as the primary bench feels somewhat dated; considering the current landscape of the field, the inclusion of more contemporary, larger-scale datasets and popular foundation models would have significantly broadened the paper's appeal and relevance. Secondly, I am concerned about the absolute performance levels reported on ImageNet. The results appear considerably lower than those achieved by general training models on the full dataset. Since the fundamental motivation of dataset distillation is to reduce data volume while preserving the original accuracy as much as possible, this substantial degradation in performance raises valid questions regarding the practical utility of the proposed method. Third, Consequently, given that the baseline performance is relatively low, the demonstrated gains against baselines in the cross-architecture settings appear less impactful than they might otherwise be.

Regarding the paper's contributions, I note a consensus among the reviewers regarding the algorithmic novelty. Reviewers gQzu, Zzhq, 98qT, and Pn8q all share the opinion that the algorithmic contribution is somewhat incremental. I specifically align with Reviewer Zzhq’s assessment that the theoretical foundation relies heavily on existing proofs from quantization theory; thus, the novelty in terms of theoretical contribution is limited, a point the authors have also acknowledged. Furthermore, I agree with Reviewer 98qT’s observation that the readability of the paper requires improvement to better communicate these ideas.

Despite these criticisms, I believe the core discovery linking dataset distillation to optimal quantization is insightful and holds value for the community. Most of the technical concerns raised by the reviewers have been adequately addressed during the rebuttal phase. Therefore, acknowledging the merit of this new perspective despite the noted limitations, I recommend acceptance for this paper.

**Reviewer Concerns:**

Most of the technical concerns raised by the reviewers have been adequately addressed during the rebuttal phase.

**Reviewer Scores:**

I believe that most reviewers will maintain their current accept ratings, and significant score increases (e.g., to very high ratings) are unlikely. With respect to Reviewer 98qT, who assigned the lowest score, it remains uncertain whether the rating will be raised, as some of the his/her concerns are, to a certain extent, reasonable. This assessment is based on the assumption of a fair evaluation process without any non-technical issues.

---

### Decision · Program_Chairs · 2026-01-26

Accept (Poster)